# A contrastive rule for meta-learning

**Nicolas Zucchet**[*]
Department of Computer Science
ETH Zürich
nzucchet@inf.ethz.ch

**Simon Schug**[*]
Institute of Neuroinformatics
University of Zürich & ETH Zürich
sschug@ethz.ch

**Johannes von Oswald**[*]
Department of Computer Science
ETH Zürich
voswaldj@ethz.ch

**Dominic Zhao**
Institute of Neuroinformatics
University of Zürich & ETH Zürich
dozhao@ethz.ch

**João Sacramento**
Institute of Neuroinformatics
University of Zürich & ETH Zürich
rjoao@ethz.ch

## Abstract

Humans and other animals are capable of improving their learning performance as they solve related tasks from a given problem domain, to the point of being able to learn from extremely limited data. While synaptic plasticity is generically thought to underlie learning in the brain, the precise neural and synaptic mechanisms by which learning processes improve through experience are not well understood. Here, we present a general-purpose, biologically-plausible meta-learning rule which estimates gradients with respect to the parameters of an underlying learning algorithm by simply running it twice. Our rule may be understood as a generalization of contrastive Hebbian learning to meta-learning and notably, it neither requires computing second derivatives nor going backwards in time, two characteristic features of previous gradient-based methods that are hard to conceive in physical neural circuits. We demonstrate the generality of our rule by applying it to two distinct models: a complex synapse with internal states which consolidate task-shared information, and a dual-system architecture in which a primary network is rapidly modulated by another one to learn the specifics of each task. For both models, our meta-learning rule matches or outperforms reference algorithms on a wide range of benchmark problems, while only using information presumed to be locally available at neurons and synapses. We corroborate these findings with a theoretical analysis of the gradient estimation error incurred by our rule.[1]

## 1   Introduction

The seminal study of Harlow [1] established that humans and non-human primates can become better at learning when presented with a series of learning tasks which share a certain common structure. To achieve this, the brain must extract and encode whichever aspects are common within a problem domain, in such a way that future learning performance is improved. This capacity, which we refer to as meta-learning, confers great evolutionary advantage to an organism over another that must

---

[*]Equal contribution; arbitrary ordering.
[1]Code available at https://github.com/smonsays/contrastive-meta-learning

face new tasks starting from *tabula rasa*. The neural and synaptic basis of this higher-order form of learning is largely unknown and theories are notably scarce [2]. The present work focuses on developing one such theory.

Formally, we define learning as the optimization of a data-dependent objective function with respect to learnable parameters, following the prevalent view in machine learning [3]. Meta-learning can be straightforwardly accommodated for in this framework by first specifying a learning algorithm through a set of meta-parameters, and then measuring post-learning performance through a meta-objective function [4–8]. Formulated as such, meta-learning corresponds to a hierarchical optimization problem, where lower-level parameters are optimized to learn the specifics of each task, and meta-parameters are adapted over tasks to improve overall learning performance.

An essential question in this framework is how to optimize meta-parameters. In current deep learning practice, meta-parameters are almost always learned by backpropagation-through-learning, an instance of backpropagation-through-time [9]. While a number of biologically-plausible designs [3, 10–13] have been developed for the standard error backpropagation algorithm for feedforward neural networks [14, 15], backpropagation-through-learning suffers from a number of issues which appear to be fundamentally difficult to overcome in biological circuits. For example, when learning involves optimizing synaptic connection weights – as it is presumed to be the case in the brain – implementing backpropagation-through-learning would entail backtracking through a sequence of synaptic changes in reverse-time order, while carrying out operations which would require knowledge of all synaptic weights to be available at a single synapse. This is clearly at odds with what is currently known about synaptic plasticity. Thus, calculating meta-parameter gradients by backpropagation is both computationally expensive, and hard to reconcile with biological constraints.

Here we present a meta-learning rule for adapting meta-parameters which does not exhibit such issues. Instead of backpropagating through a learning process, our rule estimates meta-parameter gradients by running the underlying learning algorithm twice: learning a task is followed by a second run to solve an augmented learning problem which includes the meta-objective. Our rule has a number of appealing properties: (1) it runs forward in time, making the learning rule causal; (2) implementing it only requires temporarily buffering one intermediate state; (3) it does not evaluate second derivatives, thus avoiding accessing information that is non-local to a parameter; and (4) it approximates meta-gradients as accurately as needed. Furthermore, our rule is generically applicable and it can be used to learn any meta-parameter which influences the meta-objective function.

The local and causal nature of our rule allows us to develop a theory of meta-plastic synapses, which slowly consolidate information over tasks in their internal hidden states or in their synaptic weights. We show through experiments that, when governed by our meta-learning rule, such slow adaptation processes result in improved learning performance in a variety of benchmark problems and network architectures, from deep convolutional to recurrent spiking neural networks, on both supervised and reinforcement learning paradigms. Moreover, we find that our meta-learning rule performs as well or better than reference methods, including backpropagation-through-learning, and we provide a theoretical bound for its meta-gradient estimation error which is confirmed by our experimental findings. Thus, our results demonstrate that gradient-based meta-learning is possible with local learning rules, and suggest ways by which slower synaptic processes in the brain optimize the performance of faster learning processes.

## 2   Background and problem setup

The goal of meta-learning is to improve the performance of a learning algorithm through experience. We begin by formalizing this goal as a mathematical optimization problem and outlining its solution with standard gradient-based methods. The approach we present below underlies a large body of work studying meta-learning in neural networks [e.g., 7, 16–18]. We also discuss why these standard methods may be deemed unsatisfactory as models of meta-learning in the brain.

**Problem setup.**   Formally, we wish to optimize the meta-parameters $\theta$ of an algorithm which learns to solve a given task $\tau$ by changing the parameters $\phi$ of a model. Each task is drawn from a distribution $p(\tau)$ representing the problem domain and comes with an associated loss function $L_\tau^{\text{learn}}(\phi, \theta)$, which depends on some data $D_\tau^{\text{learn}}$. The goal of learning is to minimize this loss while keeping the meta-parameters $\theta$ fixed; we denote the outcome of learning task $\tau$ by $\phi_{\theta,\tau}^*$. The subscript

$\theta$ in $\phi_{\theta,\tau}^*$ is here to emphasize that the solution of a task implicitly depends on the meta-parameters $\theta$ used during learning. Learning performance is then evaluated by measuring again a loss function $L_\tau^{\text{eval}}(\phi_{\theta,\tau}^*, \theta)$, defined on new evaluation data $D_\tau^{\text{eval}}$ from the same task. The meta-objective is this evaluation loss, averaged over tasks. Hence, we formalize meta-learning as a bilevel optimization problem, which can be compactly written as follows:

$$\min_{\theta} \; \mathbb{E}_{\tau \sim p(\tau)} \left[ L_\tau^{\text{eval}}(\phi_{\theta,\tau}^*, \theta) \right] \quad \text{s.t.} \quad \phi_{\theta,\tau}^* \in \arg\min_{\phi} L_\tau^{\text{learn}}(\phi, \theta). \tag{1}$$

In this paper, we approach problem (1) with stochastic gradient descent, which uses meta-gradient information to update meta-parameters after learning a task (or a minibatch of tasks) presented by the environment. For a given task $\tau$ we thus need to compute the meta-gradient

$$\nabla_{\theta,\tau} := \left( \frac{\mathrm{d}}{\mathrm{d}\theta} L_\tau^{\text{eval}}(\phi_{\theta,\tau}^*, \theta) \right)^\top. \tag{2}$$

The implicit dependence of $\phi_{\theta,\tau}^*$ on the meta-parameters $\theta$ complicates the computation of the meta-gradient; differentiating through the learning algorithm efficiently is a central question in gradient-based meta-learning. We next review two major known ways of doing so.

**Review of backpropagation-through-learning.** A common strategy followed in previous work [cf. 19] is to replace the solution $\phi_{\theta,\tau}^*$ to a learning task by the result $\phi_{\theta,\tau,T}$ obtained after applying a differentiable learning algorithm for $T$ time steps, not necessarily until convergence. One advantage of this formulation is that the computational graph for $\phi_{\theta,\tau,T}$ is explicitly available. Thus, backpropagation can be invoked to compute the meta-gradient $\nabla_{\theta,\tau}$, yielding what we refer to as backpropagation-through-learning. This approach is hardly biologically-plausible, as it requires storing and revisiting the parameter trajectory $\{\phi_t\}_{t=1}^T$ backwards in time, from $t = T$ to $t = 0$. Moreover, when the learning algorithm which produces $\phi_{\theta,\tau,T}$ is itself gradient-based, as it typically is in deep learning, differentiating through learning gives rise to second derivatives. These second-order terms involve cross-parameter dependencies that are difficult to resolve with local processes.

**Review of implicit differentiation.** An alternative line of methods [20–23] approaches problem (1) through the implicit function theorem [24]. This theorem provides conditions under which the meta-gradient $\nabla_{\theta,\tau}$ is well-defined, while also providing a formula for it. Over backpropagation-through-learning, this approach has the advantages that it does not require storing parameter trajectories $\{\phi_t\}_{t=1}^T$, and that it is agnostic to which algorithm is used to learn a task. However, the meta-gradient formula provided by the implicit function theorem is difficult to evaluate directly for neural network models, as it includes the inverse learning loss Hessian. This makes it hard to design biologically-plausible meta-learning algorithms based directly on the implicit meta-gradient expression. We refer to Section S2 for more details and an expanded discussion on this class of meta-learning methods.

## 3   Contrastive meta-learning

Here we present a new meta-learning rule which is generically applicable to meta-learning problems of the form (1). Our rule is gradient-following, and therefore scalable to neural network problems involving high-dimensional meta-parameters, while being simpler to conceive in biological neural circuits than the standard gradient-based methods reviewed in the previous section.

To derive our meta-learning rule we first introduce an auxiliary objective function which mixes the two levels of the bilevel optimization problem (1):

$$\mathcal{L}_\tau(\phi, \theta, \beta) = L_\tau^{\text{learn}}(\phi, \theta) + \beta L_\tau^{\text{eval}}(\phi, \theta). \tag{3}$$

We refer to $\mathcal{L}_\tau(\phi, \theta, \beta)$ as the augmented loss function. This auxiliary loss depends on a new scalar parameter $\beta \in \mathbb{R}$, which we call the nudging strength. Positive values of $\beta$ nudge learning towards the meta-objective associated with task $\tau$. Thus, we can define a family of auxiliary learning problems through the augmented loss $\mathcal{L}_\tau$ by varying the nudging strength $\beta$ away from zero. We denote the solutions to these auxiliary learning problems by

$$\phi_{\theta,\beta,\tau}^* \in \arg\min_{\phi} \mathcal{L}_\tau(\phi, \theta, \beta), \tag{4}$$

and we use $\hat{\phi}_{\theta,\beta,\tau}$ to distinguish approximate model parameters found in practice with some learning algorithm from the true minimizers $\phi^*_{\theta,\beta,\tau}$. Note that for the special case of $\beta = 0$, we recover a solution $\phi^*_{\theta,0,\tau}$ of the original learning task defined by $L^{\text{learn}}_\tau(\phi, \theta)$.

Our contrastive meta-learning rule prescribes the following change to the meta-parameters $\theta$ after encountering learning task $\tau$:

$$\Delta_{\theta,\tau} := -\frac{1}{\beta} \left( \frac{\partial \mathcal{L}_\tau}{\partial \theta}(\hat{\phi}_{\theta,\beta,\tau}, \theta, \beta) - \frac{\partial \mathcal{L}_\tau}{\partial \theta}(\hat{\phi}_{\theta,0,\tau}, \theta, 0) \right)^\top. \tag{5}$$

This rule contrasts information over two model parameter settings, $\hat{\phi}_{\theta,0,\tau}$ and $\hat{\phi}_{\theta,\beta,\tau}$; it may be understood as a generalization to meta-learning of a classical recurrent neural network learning algorithm known as contrastive Hebbian learning [25–29]. Intuitively, as we compute the solution to the augmented learning problem with $\beta > 0$, we nudge our learning algorithm towards a parameter setting $\hat{\phi}_{\theta,\beta,\tau}$ that would have been better in terms of the meta-objective — that we wish our algorithm had actually reached, without needing the meta-objective to influence the learning process.

Our rule implements meta-learning by gradient descent when the learning solutions $\hat{\phi}_{\theta,0,\tau}$ and $\hat{\phi}_{\theta,\beta,\tau}$ are exact and as $\beta \to 0$. This important property can be shown by invoking the equilibrium propagation theorem [29, 30] discovered and proved by Scellier and Bengio; we restate this result and present the technical conditions for applying it to meta-learning in Section S1. Critically, $\Delta_{\theta,\tau}$ estimates the meta-gradient $\nabla_{\theta,\tau}$ using only partial derivative information and without ever directly calculating the total derivative in (2). Depending on the model, partial derivatives of the augmented loss $\mathcal{L}_\tau$ may be easy to calculate analytically and implement, or they may require dedicated neural circuits for their evaluation; we return to this point in the next section.

We recall that the two points $\hat{\phi}_{\theta,0,\tau}$ and $\hat{\phi}_{\theta,\beta,\tau}$ which appear in (5) respectively correspond to approximate solutions of the original and the augmented learning problems. Thus, the information required to implement our rule can be collected causally by invoking the learning algorithm for a second time, after the actual task has been learned, while buffering information across the two runs. In contrast to backpropagation-through-learning, this process runs forward in time, it only requires keeping a single intermediate state in short-term memory, and it is entirely agnostic to which underlying learning algorithm is used. Moreover, as we will show in the theoretical results, its precision can be varied; the same rule can produce both coarse- and fine-grained meta-gradient estimates as needed, by varying the amount of resources spent in learning and by controlling the nudging strength $\beta$.

## 4 Models

In the previous section, our contrastive meta-learning rule was presented in its general form. We now describe two concrete neural models that provide complementary views on how meta-learning could be conceived in the brain. We study the specific meta-learning rules arising from the application of the update (5) to each case and discuss their implementation with biological neural circuitry.

### 4.1 Synaptic consolidation as meta-learning

We first use our general contrastive meta-learning rule (5) to derive meta-plasticity rules for a complex synapse model which has been featured in prior meta-learning [22, 31] and continual learning [32, 33] work. Biological synapses are complex devices which comprise components that adapt at multiple time scales. Beyond changes induced by standard long-term potentiation and depression protocols lasting minutes to several hours, synapses exhibit activity-dependent plasticity at much longer time scales [34–36]. While previous work has focused on characterizing memory retention in more realistic synapse models, here we study how such slow synaptic consolidation processes may support fast future learning through our contrastive meta-learning rule.

In the model we consider, besides a synaptic weight $\phi$ which influences postsynaptic activity, each synapse has an internal consolidated state $\omega$ towards which the weight is attracted whenever the synapse changes. We further allow the attraction strength $\lambda$ to vary over synapses; its reciprocal $\lambda^{-1}$ plays a role similar to a learning rate. For this model the meta-parameters are therefore $\theta = \{\lambda, \omega\}$.

We model the interaction between these three components through a quadratic function, which is added to the task-specific learning loss $l_\tau^{\text{learn}}(\phi)$:

$$L_\tau^{\text{learn}}(\phi, \theta) = l_\tau^{\text{learn}}(\phi) + \frac{1}{2} \sum_{i=1}^{|\phi|} \lambda_i (\omega_i - \phi_i)^2. \tag{6}$$

In machine learning terms, we regularize the learning loss with a quadratic regularizer. On the other hand, the evaluation loss function $L_\tau^{\text{eval}}(\phi)$ depends only on the synaptic weights $\phi$ such that the meta-parameters $\theta$ only influence learning, not prediction.

The partial derivatives which appear in our contrastive meta-learning rule (5) can be analytically obtained for this synaptic model. A calculation yields the meta-plasticity rules

$$\Delta_{\omega,\tau} = \frac{\lambda}{\beta} \left( \hat{\phi}_{\theta,\beta,\tau} - \hat{\phi}_{\theta,0,\tau} \right) \quad \text{and} \quad \Delta_{\lambda,\tau} = \frac{1}{2\beta} \left[ (\hat{\phi}_{\theta,0,\tau} - \omega)^2 - (\hat{\phi}_{\theta,\beta,\tau} - \omega)^2 \right], \tag{7}$$

where all operations are carried out elementwise. Contrastive meta-learning thus offers a principled way to slowly (over learning tasks) consolidate information in the internal states of complex synapses to improve future learning performance. Critically, it leads to meta-plasticity rules that are entirely local to a synapse and are independent of the method used to learn. Our meta-plasticity rules can thus be flexibly applied to improve the performance of any learning algorithm, including a host of biologically-plausible learning rules, from precise neuron-specific error backpropagation circuits [37, 38] to stochastic perturbation reinforcement rules [39]. The only requirement our theory makes is that learning corresponds to the optimization of an objective.

## 4.2 Learning by top-down modulation

The second model that we consider is inspired by the modulatory role that is attributed to top-down inputs from higher- to lower-order brain areas. Such modulatory inputs often feature in neural theories of attention and contextual processing [40–42]. Here, we explore the possibility that they subserve fast learning of new tasks. We incorporate this insight into a simple meta-learning model, where learning a task $\tau$ corresponds to finding the right pattern of task-specific modulation $\phi_{\theta,\tau}^*$, and meta-learning corresponds to changing synaptic weights $\theta$. Unlike in the complex synapse model presented in the previous section, here we interpret the task-specific parameters $\phi_\tau$ as patterns of neural activity, not synaptic weights. This implies that, if meta-learning succeeds, it becomes possible to learn new tasks on the fast neural time scale without evoking synaptic plasticity.

More concretely, we take as modulatory inputs a multiplicative gain $g$ and an adaptive threshold $b$ per neuron, as done in previous work [43, 44]. Rapid (input-dependent) multiplicative and additive modulation of the sensitivity of the neural input-output response curve $\sigma(x)$ is typically observed in cortical neurons [45]. There exist a number of biophysical mechanisms which allow top-down inputs to modulate $\sigma(x)$ [e.g., 46]. Assuming a simple linear-threshold neuron model with weights $\theta$, this yields the response $\sigma(x) = g(\theta \cdot x - b)_+$ to some input $x$, where $(\cdot)_+$ denotes the positive-part operation. In this model, there are only few learnable parameters $\phi = \{g, b\}$, as they scale with the number of neurons and not with the number of synaptic connections.

We apply contrastive meta-learning to this model by changing synaptic weights $\theta$ according to our rule (5). For this model, partial derivatives of the augmented loss function correspond to the usual derivatives with respect to model parameters that are routinely evaluated to learn deep neural networks; our rule simply asks to compute them twice. We therefore build upon existing theories of learning by backpropagation-of-error in the brain and assume that some mechanism for neuron-specific spatial error backpropagation is available, for example via prediction error neural subpopulations [37] or dendritic error representations [11, 38, 47], or by invoking equilibrium propagation again [29].

## 5 Theoretical and experimental analyses

In the following, we theoretically analyze the approximation error incurred by our contrastive meta-learning rule before empirically testing it on a suite of meta-learning problems. The objective of our experiments is twofold. First, we aim to confirm our theoretical results and demonstrate the performance of contrastive meta-learning on standard machine learning benchmarks. Second, we want to illustrate the generality of our approach by applying it to various supervised and reinforcement meta-learning problems as well as to a more biologically realistic neuron and plasticity model.

## 5.1 Theoretical analysis of the meta-gradient approximation error

The contrastive meta-learning rule (5) only provides an approximation to the meta-gradient. This approximation can be improved by refining the two learning solutions $\hat{\phi}_{\theta,0,\tau}$ and $\hat{\phi}_{\theta,\beta,\tau}$ through additional computation or by using a better learning algorithm, and by decreasing the nudging strength $\beta$, as prescribed by the equilibrium propagation theorem. In Theorem 1, we theoretically analyze how the meta-gradient estimate (5) benefits from such improvements (see Fig. 1A for a visualization of the result, Section S3 for a proof and empirical verification of our theoretical results). We find that the refinement of the learning solutions must be coupled to a decrease in $\beta$: too small $\beta$ greatly detracts from the quality of the meta-gradient estimate when the solutions are not improved, while better approximations are inefficient if $\beta$ is not decreased accordingly.

**Theorem 1** (Informal). *Let $\beta > 0$ and $\delta$ be such that $\|\hat{\phi}_{\theta,0,\tau} - \phi^*_{\theta,0,\tau}\| \leq \delta$ and $\|\hat{\phi}_{\theta,\beta,\tau} - \phi^*_{\theta,\beta,\tau}\| \leq \delta$. Then, under regularity and convexity assumptions, there exists a constant $C$ such that*

$$\|-\Delta_{\theta,\tau} - \nabla_{\theta,\tau}\| \leq C \left( \frac{1+\beta}{\beta}\delta + \frac{\beta}{1+\beta} \right) =: \mathcal{B}(\delta, \beta).$$

## 5.2 Contrastive meta-learning is a high-performance meta-optimization algorithm

As a first set of experiments, we study a supervised meta-optimization problem based on the entire CIFAR-10 image dataset [48]. In these experiments the goal is to meta-learn a set of hyperparameters (meta-parameters) such that generalization performance improves. This problem is a common testbed for assessing the ability of a meta-learning algorithm to optimize a given meta-objective [23]; it can be thought of as a limiting case of full meta-learning, as there are learnable meta-parameters, but only one task. As the meta-objective we take the cross-entropy loss $l$ evaluated on a held-out dataset $D^{\text{eval}}$: $L^{\text{eval}}(\phi) = \frac{1}{|D^{\text{eval}}|} \sum_{(x,y) \in D^{\text{eval}}} l(x, y, \phi)$, where $x$ is an image input and $y$ its label. We equip a convolutional deep neural network with our synaptic model (6), meta-learning only the per-synapse regularization strength $\lambda$, keeping $\omega$ fixed at zero: $L^{\text{learn}}(\phi, \lambda) = \frac{1}{|D^{\text{learn}}|} \sum_{(x,y) \in D^{\text{learn}}} l(x, y, \phi) + \frac{1}{2} \sum_{i=1}^{|\phi|} \lambda_i \phi_i^2$. We learn the weights $\phi$ by stochastic gradient descent paired with backpropagation. Additional details and analyses may be found in Section S4.1.

We benchmark our meta-plasticity rule (7) against implicit gradient-based meta-learning methods, which are considered state-of-the-art for this type of problem [23] (see Section S2 for a review). More concretely, recurrent backpropagation (RBP [49, 50]; also known as the Neumann series approximation [23, 51]) and the conjugate gradient method (CG) [21, 52] correspond to two different numerical schemes for calculating the meta-gradient; T1-T2 [53] is an approximate method which neglects complicated terms, thus introducing a non-reducible bias in the meta-gradient estimate. Critically, unlike our contrastive meta-learning rule (CML), this method offers no control over the meta-gradient error.

Table 1: Meta-learning a per-synapse regularization strength meta-parameter (cf. Section 4.1) on CIFAR-10. Average accuracies (acc.) $\pm$ s.e.m. over 10 seeds.

| Method | Evaluation acc. (%) | Test acc. (%) |
|---|---|---|
| T1-T2 | $64.77^{\pm0.40}$ | $62.57^{\pm0.31}$ |
| CG | $57.65^{\pm1.51}$ | $57.51^{\pm0.98}$ |
| RBP | $64.92^{\pm1.32}$ | $62.14^{\pm0.97}$ |
| CML | $74.43^{\pm0.53}$ | $66.94^{\pm0.25}$ |
| No meta | $60.06^{\pm0.37}$ | $60.13^{\pm0.38}$ |
| TBPTL | $73.17^{\pm0.27}$ | $65.35^{\pm0.36}$ |

We find that our meta-learning rule outperforms all three baseline implicit differentiation methods in terms of both evaluation-set and actual generalization (test-set) performance, cf. Tab. 1. As a side result, we confirm the instability of CG in deep learning reported in ref. [51, 54]. We note that the hyperparameters of all four methods were independently and carefully set (cf. Section S4.1). These strong results on a modern deep learning benchmark, involving stochastic approximate learning, demonstrate that contrastive meta-learning is a scalable, highly effective meta-optimization algorithm. Moreover, Theorem 1 is in excellent qualitative agreement with our experiments, cf. Fig. 1.

To further contextualize our findings, we provide results for training the same network without meta-learning, where we performed a conventional hyperparameter search over a scalar regularization strength hyperparameter shared by all synapses. This simple approach yields only a moderate evaluation and test accuracy.

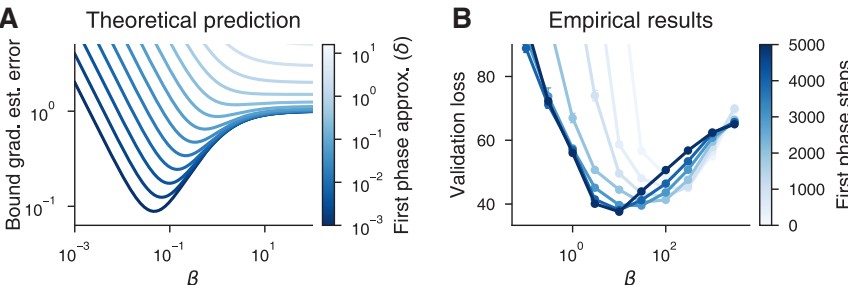

Figure 1: (A) Visualization of the theoretical bound $\mathcal{B}$ on the meta-gradient estimation error from Theorem 1 as a function of the nudging strength $\beta$. Better approximations of the solutions (smaller $\delta$) improve the quality of the meta-gradient, as they enable using smaller values of $\beta$. (B) Confirmation of the qualitative findings of the theory on deep learning experiments. We show results for a hyperparameter meta-learning problem, where a per-synapse regularization strength is meta-learned (cf. Section 4.1) on CIFAR-10 with rule (7). The validation loss is a proxy for the quality of the gradient and the number of steps in the first phase is a proxy for $-\log\delta$.

As all methods incur numerical errors when computing the meta-gradient, a comparison to using the analytical solution for the meta-gradient would be desirable. Since this is intractable in this case and running full backpropagation-through-learning requires too much memory, we evaluate truncated backpropagation-through-learning (TBPTL) with the maximal truncation window we can fit on a single graphics processing unit (in our case 200 out of 5000 steps). The resulting evaluation accuracy and test accuracy outperform other implicit gradient-based meta-learning methods but are still surpassed by our method.

## 5.3 Contrastive meta-learning enables visual few-shot learning

The ability to learn new object classes based on only a few examples is a hallmark of human intelligence [55] and a prime application of meta-learning. We test whether our contrastive meta-learning rule is able to turn into a few-shot learner a standard visual system, a convolutional deep neural network learned by gradient descent and error backpropagation. Furthermore, we ask how our contrastive meta-learning rule fares against other gradient-based meta-learning algorithms which rely on backpropagation-through-learning and implicit differentiation to compute gradients. To that end, we focus on two widely-studied few-shot image classification problems based on miniImageNet [56] and the Omniglot [57] datasets. To further facilitate comparisons, we reproduce exactly the experimental setup of ref. [18], which has been adopted in a large number of studies.

Briefly, during meta-learning, $N$-way $K$-shot tasks are created on-the-fly by sampling $N$ classes at random from a fixed pool of classes, and then splitting the data into task-specific learning $D_\tau^{\text{learn}}$ (with $K$ examples per class for learning) and evaluation $D_\tau^{\text{eval}}$ sets, used to define the corresponding loss functions $L_\tau^{\text{learn}}$ and $L_\tau^{\text{eval}}$. The meta-objective is then simply the task-averaged evaluation loss, measured after learning. The performance of the learning algorithm is tested on new tasks consisting of classes that were not seen during meta-learning. We provide all experimental details in Section S4.2.

As reference methods, we compare against the well-known model-agnostic meta-learning (MAML) algorithm [18], which relies on backpropagation-through-learning to meta-learn an initial set of weights, starting from which a few gradient steps should succeed; this is conceptually similar to meta-learning the consolidated state $\omega$ of our complex synapses. We also include results obtained with its first-order approximation FOMAML (as well as a closely related algorithm known as Reptile [58]), which, like the T1-T2 algorithm of the previous section, excludes all second-order terms from the meta-gradient estimate to simplify the update, at the expense of introducing a bias. Finally,

Table 2: One-shot miniImageNet learning. Averages over 5 seeds $\pm$ std.

| Method | Test acc. (%) |
|---|---|
| MAML [18] | $48.70^{\pm 1.84}$ |
| FOMAML [18] | $48.07^{\pm 1.75}$ |
| Synaptic | $48.43^{\pm 0.43}$ |
| Modulatory | $49.80^{\pm 0.40}$ |

we compare to the implicit MAML (iMAML) algorithm [22], which corresponds exactly to meta-learning our consolidated synaptic state $\omega$, but with implicit differentiation methods.

When applied to the problem domain of miniImageNet one-shot learning tasks, the performance of all meta-learning algorithms we consider here is closely clustered together, cf. Tab. 2. In particular, meta-learning the consolidated states $\omega$ of our complex synapses with implicit differentiation (iMAML) or our local update (7) leads to comparable performance. Interestingly, we further find that miniImageNet one-shot learning performance is significantly improved

Table 3: Omniglot character few-shot learning. Test set classification accuracy (%) averaged over 5 seeds $\pm$ std.

| Method | 20-way 1-shot | 20-way 5-shot |
|---|---|---|
| MAML [18] | $95.8^{\pm 0.3}$ | $98.9^{\pm 0.2}$ |
| FOMAML [18] | $89.4^{\pm 0.5}$ | $97.9^{\pm 0.1}$ |
| Reptile [58] | $89.43^{\pm 0.14}$ | $97.12^{\pm 0.32}$ |
| iMAML [22] | $94.46^{\pm 0.42}$ | $98.69^{\pm 0.1}$ |
| CML (synaptic) | $94.16^{\pm 0.12}$ | $98.06^{\pm 0.26}$ |
| CML (modulatory) | $94.24^{\pm 0.39}$ | $98.60^{\pm 0.27}$ |

when using the modulatory model described in Section 4.2, despite the low dimensionality of the task-specific variable $\phi$. This is in line with other results suggesting that highly efficient visual learning of new categories may be possible without necessarily engaging synaptic plasticity [43]. On Omniglot (see Section S4.2 for additional variants), the situation is comparable, except that on its 20-way 1-shot variant, the performance gap between first- and second-order methods widens. In line with our theory, our contrastive meta-learning rule performs close to (second-order) implicit differentiation, showing that despite its simplicity and locality our rule is able to accurately estimate meta-gradients.

## 5.4 Contrastive meta-learning enables meta-plasticity in a recurrent spiking network

For the experiments described on the previous sections we used simple artificial neuron models and backpropagation-of-error to learn. We now move closer to a biological neuron and plasticity model and consider meta-learning in a recurrently-connected neural network of leaky integrate-and-fire neurons with plastic synapses. We study a simple few-shot regression problem [18], where the aim is to quickly learn to approximate sinusoidal functions which differ in their phase and amplitude (for additional details see Section S4.3). For each task, we measure the mean squared error on 10 samples for the learning loss and 10 samples for the evaluation loss. We implement synaptic plasticity using the local e-prop rule [59] and use a population of 100 Poisson neurons to encode inputs, see Fig. 2A. As our contrastive meta-learning rule (5) is agnostic to the specifics of the learning process, we can augment the model with our synaptic consolidation model and apply the meta-plasticity rules derived in (7). Fig. 2B illustrates how the learning process improves with increasing number of tasks encountered, eventually consolidating a sinusoidal prior that can be quickly adapted to the specifics of a task from few examples, cf. Fig. 2C.

We compare our method to a standard baseline where updates are computed by backpropagating through the synaptic plasticity process (backpropagation-through-learning; BPTL) using surrogate gradients to handle spiking nonlinearities [60] similar to previous work on spiking neuron meta-learning [61]. Since full BPTL requires reducing the number of learning steps compared to our method due to memory constraints, we also include

Table 4: Few-shot learning of sinusoidal functions with a recurrent spiking neural network. Avg. mean squared error (MSE) over 10 seeds $\pm$ s.e.m.

| Method | Validation MSE | Test MSE |
|---|---|---|
| BPTL + BPTT | $0.17^{\pm 0.01}$ | $0.41^{\pm 0.10}$ |
| BPTL + e-prop | $0.52^{\pm 0.05}$ | $0.72^{\pm 0.08}$ |
| TBPTL + e-prop | $0.27^{\pm 0.07}$ | $0.50^{\pm 0.11}$ |
| CML + e-prop | $0.23^{\pm 0.04}$ | $0.23^{\pm 0.04}$ |

TBPTL with the same number of 500 learning steps and a truncation window of 100 steps. In both cases, we find competitive performance for our method, see Tab. 4.

## 5.5 Contrastive meta-learning improves reward-based learning

Finally, we demonstrate how contrastive meta-learning can be applied in the challenging setting of reward-based learning, second nature to most animals. Reward-based learning clearly demonstrates hallmarks of meta-learning as animals are capable of flexibly remapping reward representations when task contingencies change [62, 63]. Inspired by this, we aim to meta-learn a value function on a

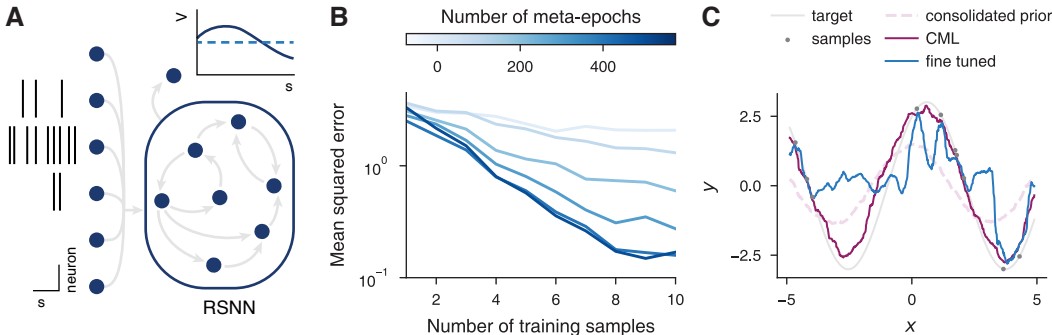

Figure 2: (A) A network of recurrently-connected leaky-integrate and fire neurons is tasked with learning sinusoids on an input encoding of Poisson spike trains. Its prediction is the voltage of the output neuron averaged over time. (B) Learning performance from few examples measured as the mean squared error on evaluation examples during a learning episode improves as more tasks are encountered over the course of meta-learning. (C) Meta-plasticity encodes information on the consolidated synaptic component (dashed) which results in improved learning performance (purple), compared to a naive network learning from scratch (blue).

family of reward-based learning tasks that can be quickly adapted to predict the expected reward of the actions available to the agent in a particular task.

Specifically, we consider the wheel bandit problem introduced by [64] with the meta-learning setup previously studied in refs. [65, 66]. On each task, an agent is presented with a sequence of context coordinates randomly drawn from a unit circle for each of which it has to choose among 5 actions to receive a stochastic reward. Hidden to the agent, a task-specific radius $\delta$ tiles the context space into a low- and a high-reward region depending on which the optimal action to take changes (see Section S4.4).

The goal of meta-learning is to discern the general structure of the low- and high-reward region across tasks whereas the goal of learning becomes to identify the task-specific radius $\delta$ of the current task. During meta-learning, we randomly sample tasks $\delta \sim \mathcal{U}(0, 1)$ and generate a dataset by choosing actions randomly. Data from each task is split into training and evaluation data, ef-

Table 5: Cumulative regret on the wheel bandit problem for different $\delta$. Values normalized by the cumulative regret of a uniformly random agent. Avgs. over 50 seeds $\pm$ s.e.m.

| $\delta$ | 0.5 | 0.9 | 0.99 |
| --- | --- | --- | --- |
| NeuralLinear [64] | $0.95^{\pm 0.02}$ | $4.65^{\pm 0.18}$ | $49.63^{\pm 2.41}$ |
| MAML | $0.45^{\pm 0.01}$ | $1.02^{\pm 0.76}$ | $15.21^{\pm 1.69}$ |
| CML (synaptic) | $0.40^{\pm 0.02}$ | $0.82^{\pm 0.02}$ | $12.27^{\pm 1.02}$ |
| CML (modulatory) | $0.42^{\pm 0.01}$ | $1.83^{\pm 0.11}$ | $16.46^{\pm 1.80}$ |

fectively creating a sparse regression problem where only the outcome of a randomly chosen action can be observed for a particular context. After meta-learning, we evaluate the cumulative regret obtained by an agent that chooses his actions greedily with respect to its predicted rewards and adapts its fast parameters on the observed context, action, reward triplets stored in a replay buffer.

We use both our synaptic consolidation and modulatory network models to meta-learn the value function using our contrastive rule. We compare our two models to MAML and the non-meta-learned baseline, NeuralLinear, from ref. [64], which performed among the best in their large-scale comparison. Tab. 5 shows the cumulative regret obtained on different task parametrizations $\delta$ in the online evaluation after meta-learning (extended table in Section S4.4). Meta-learning clearly improves upon the non-meta-learned baseline with both our models performing comparably to MAML. This improvement is more pronounced for tasks with larger $\delta$ within which it is more difficult to discover the high-reward region.

## 6   Discussion

We have presented a general-purpose meta-learning rule which allows estimating meta-gradients from local information only, and we have demonstrated its versatility studying two neural models on a range of meta-learning problems. The competitive performance we observed suggests that contrastive meta-

learning is a worthy contender to biologically-implausible machine learning algorithms – especially for problems involving long learning trajectories, as demonstrated by the strong results on supervised meta-optimization. At its core, our method relies on contrasting the outcome of two different learning episodes. Despite its conceptual simplicity this requires complex synaptic machinery which is able to buffer these outcomes in a way accessible to synaptic consolidation.

According to our top-down modulation model the goal of synaptic plasticity in primary brain areas is *not* to learn a specific task, in contrast to more traditional theories of learning. Instead, we postulate that the goal of synaptic plasticity is to make it possible to learn any given task by modulating the sensitivity of primary-area neurons in a task-dependent manner. This view is consistent with the experimental findings of Fritz et al. [67], who observed the rapid formation of task-dependent receptive fields in the primary auditory cortex of ferrets, as the animals learned several tasks, presumably due to changes in top-down signals originating in frontal cortex. Together with the strong results of the modulatory model in the challenging setting of visual one-shot learning and recent studies in continual learning problems [68–71] this shows the practical effectiveness of task-dependent modulation. Complementary to the interaction of the frontal cortex with primary cortical areas, the prefrontal cortex might similarly modulate the striatum during reward-based learning. Whereas classical dopamine-based learning posits that reward prediction errors are used subcortically to learn the reward structure of a task, recent work has demonstrated that reward can similarly affect prefrontal representations to quickly infer the current task identity and switch the context provided to the striatum [72]. More broadly, viewing synaptic plasticity as meta-learning is also consistent with recent modeling work casting the prefrontal cortex as a meta-reinforcement learning system [73].

Reflecting on how our meta-learning rule can be implemented in the brain, we conjecture that the hippocampal formation plays a central role in coordinating the two phases as well as creating the augmented learning problem. First, some mechanism must signal that a switch from learning problem to augmented learning problem has occurred, corresponding to the sign switch in our rule (5). We argue that the hippocampus is well positioned for signaling such a switch to cortical synapses. A recent experimental study shows that the hippocampus is at least able to control cortical synaptic consolidation [74] but further evidence would be needed to support our hypothesis.

Second, we conjecture that the creation of the augmented learning problem at the heart of our meta-gradient estimation algorithm might itself critically rely on the hippocampus. In all our experiments, this second learning problem consisted simply of new data, presented to the learning algorithm to evaluate how well learning went. Transferring additional data into cortical networks, putatively during sleep and wakeful rest, fits well with the role that is classically attributed to the hippocampus in systems consolidation and complementary learning systems theories [75, 76]. We thus speculate that the hippocampus 'prescribes' additional learning problems to the cortex, which serve the purpose of testing its generalization performance. By showing that a second 'sleep' learning phase enables meta-learning with simple plasticity rules, our results lend further credit to complementary learning systems theory, as well as to the hypothesis that dreams have evolved to assist generalization [77].

Lastly, this view of the cortex as a contrastive meta-learning system aided by the hippocampus may also help elucidate how the brain learns from an endless, non-stationary stream of data. Current artificial neural networks notoriously struggle to strike a balance between learning new knowledge and retaining old one in such continual learning problems, in particular when the data are not independent and identically distributed nor structured into clearly delineated tasks [78]. Interestingly, recent investigations have shown that meta-learning can greatly improve continual learning performance [79–83]. While details vary, the essence of these methods is to blend in past (replay) data with new data in a meta-objective function. This amounts to a different instantiation of our bilevel optimization problem (1), resulting in an augmented learning problem in which past and present data are intermixed, for which the hippocampus would again appear to be ideally positioned.

## Acknowledgments and Disclosure of Funding

This research was supported by an Ambizione grant (PZ00P3_186027) from the Swiss National Science Foundation and an ETH Research Grant (ETH-23 21-1) awarded to João Sacramento. Johannes von Oswald is funded by the Swiss Data Science Center (J.v.O. P18-03). We thank Angelika Steger, Benjamin Scellier, Greg Wayne, Abhishek Banerjee, Blake A. Richards, Nicol Harper, Thomas Akam, Mohamady El-Gaby, Rafal Bogacz, Giacomo Indiveri, Jean-Pascal Pfister, Mark van Rossum, Maciej Wołczyk, Seijin Kobayashi and Alexander Meulemans for discussions and feedback, and Charlotte Frenkel for assistance in our implementation of e-prop.

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
