**Supplementary Materials**

# A contrastive rule for meta-learning

**Nicolas Zucchet**[*], **Simon Schug**[*], **Johannes von Oswald**[*], **Dominic Zhao, João Sacramento**

## Table of Contents

## S1 Derivation of the contrastive meta-learning rule

Our contrastive meta-learning rule relies on the equilibrium propagation theorem [29, 30]. We review this result and how we use it to derive the different instances of our rule.

### S1.1 Equilibrium propagation theorem

First, we restate the equilibrium propagation theorem as presented in Scellier [30]. Recall the definition of the augmented loss

$$\mathcal{L}(\phi, \theta, \beta) = L^{\text{learn}}(\phi, \theta) + \beta L^{\text{eval}}(\phi, \theta). \tag{8}$$

Note that compared to the main text, we omit the subscript $\tau$ for conciseness. Given the augmented loss, the equilibrium propagation theorem states the following:

**Theorem S1** (Equilibrium propagation). *Let $L^{\text{learn}}$ and $L^{\text{eval}}$ be two twice continuously differentiable functions. Let $\phi^*$ be a fixed point of $\mathcal{L}(\,\cdot\,, \bar{\theta}, \bar{\beta})$, i.e.*

$$\frac{\partial \mathcal{L}}{\partial \phi}(\phi^*, \bar{\theta}, \bar{\beta}) = 0,$$

*such that $\partial_\phi^2 \mathcal{L}(\phi^*, \bar{\theta}, \bar{\beta})$ is invertible. Then, there exists a neighborhood of $(\bar{\theta}, \bar{\beta})$ and a continuously differentiable function $(\theta, \beta) \mapsto \phi^*_{\theta,\beta}$ such that $\phi^*_{\bar{\theta},\bar{\beta}} = \phi^*$ and for every $(\theta, \beta)$ in this neighborhood*

$$\frac{\partial \mathcal{L}}{\partial \phi}(\phi^*_{\theta,\beta}, \theta, \beta) = 0.$$

*Furthermore,*

$$\frac{\mathrm{d}}{\mathrm{d}\theta}\frac{\partial \mathcal{L}}{\partial \beta}\left(\phi^*_{\theta,\beta}, \theta, \beta\right) = \frac{\mathrm{d}}{\mathrm{d}\beta}\frac{\partial \mathcal{L}}{\partial \theta}\left(\phi^*_{\theta,\beta}, \theta, \beta\right)^\top.$$

*Proof.* The first point follows from the implicit function theorem [24]. Let $(\theta, \beta)$ be in a neighborhood of $(\bar{\theta}, \bar{\beta})$ in which $\phi^*_{\theta,\beta}$ is differentiable.

The symmetry of second order derivatives of a scalar function implies that

$$\frac{\mathrm{d}}{\mathrm{d}\theta}\frac{\mathrm{d}}{\mathrm{d}\beta}\mathcal{L}\left(\phi^*_{\theta,\beta}, \theta, \beta\right) = \frac{\mathrm{d}}{\mathrm{d}\beta}\frac{\mathrm{d}}{\mathrm{d}\theta}\mathcal{L}\left(\phi^*_{\theta,\beta}, \theta, \beta\right)^\top. \tag{9}$$

We then simplify the two sides of the equation. First, we look at the left-hand side and simplify $\mathrm{d}_\beta \mathcal{L}(\phi^*_{\theta,\beta}, \theta, \beta)$ using the chain rule and the fixed point condition

$$\begin{aligned}
\frac{\mathrm{d}}{\mathrm{d}\beta}\mathcal{L}(\phi^*_{\theta,\beta}, \theta, \beta) &= \frac{\partial \mathcal{L}}{\partial \beta}(\phi^*_{\theta,\beta}, \theta, \beta) + \frac{\partial \mathcal{L}}{\partial \phi}(\phi^*_{\theta,\beta}, \theta, \beta)\frac{\mathrm{d}\phi^*_{\theta,\beta}}{\mathrm{d}\beta} \\
&= \frac{\partial \mathcal{L}}{\partial \beta}(\phi^*_{\theta,\beta}, \theta, \beta).
\end{aligned} \tag{10}$$

Similarly, the $\mathrm{d}_\theta \mathcal{L}(\phi^*_{\theta,\beta}, \theta, \beta)$ term on the right-hand side is equal to $\partial_\theta \mathcal{L}(\phi^*_{\theta,\beta}, \theta, \beta)$ and we obtain the required result[2]. □

### S1.2 The contrastive meta-learning rule

Equilibrium propagation can be used to compute the gradient associated with the bilevel optimization problem studied in this paper

$$\min_\theta \ L^{\text{eval}}(\phi^*_\theta) \quad \text{s.t.} \ \phi^*_\theta \in \arg\min_\phi L^{\text{learn}}(\phi, \theta). \tag{11}$$

To do so, we first characterize $\phi^*_\theta$ through the stationarity condition

$$\frac{\partial L^{\text{learn}}}{\partial \phi}(\phi^*_\theta, \theta) = 0. \tag{12}$$

---

[2]Note that we use $\partial$ to denote partial derivatives and $\mathrm{d}$ to denote total derivatives.

As $L^{\mathrm{learn}}(\phi, \theta) = \mathcal{L}(\phi, \theta, 0)$ we can define an implicit function $\phi^*_{\theta,\beta}$ if $\partial^2_\phi L^{\mathrm{learn}}(\phi^*_\theta, \theta)$ is invertible for which $\phi^*_{\theta,0} = \phi^*_\theta$, and that satisfies, for $\beta$ close to 0,

$$\frac{\partial \mathcal{L}}{\partial \phi}(\phi^*_{\theta,\beta}, \theta, \beta) = 0. \tag{13}$$

The gradient associated with the bilevel optimization problem (11) is then equal to

$$\nabla_\theta := \left( \frac{\mathrm{d}}{\mathrm{d}\theta} L^{\mathrm{eval}}(\phi^*_\theta, \theta) \right)^\top = \frac{\mathrm{d}}{\mathrm{d}\theta} \frac{\partial \mathcal{L}}{\partial \beta}(\phi^*_{\theta,\beta}, \theta, \beta) \Big|_{\beta=0}^\top = \frac{\mathrm{d}}{\mathrm{d}\beta} \frac{\partial \mathcal{L}}{\partial \theta}(\phi^*_{\theta,\beta}, \theta, \beta) \Big|_{\beta=0}. \tag{14}$$

Since $\beta$ is a scalar, we can use finite difference methods to efficiently estimate

$$\Delta\theta = -\widehat{\nabla}_\theta = -\frac{1}{\beta}\left( \frac{\partial \mathcal{L}}{\partial \theta}(\hat{\phi}_\beta, \theta, \beta) - \frac{\partial \mathcal{L}}{\partial \theta}(\hat{\phi}_0, \theta, 0) \right)^\top, \tag{15}$$

in which $\hat{\phi}_0$ and $\hat{\phi}_\beta$ denote the estimates of $\phi^*_{\theta,0}$ and $\phi^*_{\theta,\beta}$. If those estimates are exact, we are guaranteed that the update converges to the true gradient. In some of our experiments, we use a more sophisticated center difference approximation similar to [87], that is

$$\Delta\theta^{\mathrm{sym}} = -\widehat{\nabla}_\theta^{\mathrm{sym}} = -\frac{1}{2\beta}\left( \frac{\partial \mathcal{L}}{\partial \theta}(\hat{\phi}_\beta, \theta, \beta) - \frac{\partial \mathcal{L}}{\partial \theta}(\hat{\phi}_{-\beta}, \theta, -\beta) \right)^\top. \tag{16}$$

We refer to (16) as the symmetric variant of our contrastive rule. When the estimates for the fixed points are exact, it reduces the meta-gradient estimation bias from $O(\beta)$ for the forward difference above to $O(\beta^2)$ at the expense of having to run a third phase.

### S1.3   Application to the complex synapse model

We can now derive the meta-learning rules for the complex synapse model of Section 4.1. Recall that

$$L^{\mathrm{learn}}(\phi, \theta) = l^{\mathrm{learn}}(\phi) + \frac{1}{2}\sum_{i=1}^{|\phi|} \lambda_i(\omega_i - \phi_i)^2 \tag{17}$$

and

$$L^{\mathrm{eval}}(\phi) = l^{\mathrm{eval}}(\phi), \tag{18}$$

where $l^{\mathrm{eval}}(\phi)$ and $l^{\mathrm{learn}}(\phi)$ are two data-dependent loss functions.

For the complex synapse model, only the learning loss depends on the meta-parameters, hence $\partial_\theta \mathcal{L} = \partial_\theta L^{\mathrm{learn}}$ and

$$\begin{aligned} \frac{\partial \mathcal{L}}{\partial \omega}(\phi, \theta, \beta) &= \lambda(\omega - \phi) \\ \frac{\partial \mathcal{L}}{\partial \lambda}(\phi, \theta, \beta) &= \frac{1}{2}(\omega - \phi)^2, \end{aligned} \tag{19}$$

where all the operations are carried out elementwise. Plugging the last equation in the contrastive update (15) yields

$$\begin{aligned} \Delta\omega &= -\frac{\lambda}{\beta}\left( (\omega - \hat{\phi}_\beta) - (\omega - \hat{\phi}_0) \right) = \frac{\lambda}{\beta}\left( \hat{\phi}_\beta - \hat{\phi}_0 \right) \\ \Delta\lambda &= -\frac{1}{2\beta}\left( (\omega - \hat{\phi}_\beta)^2 - (\omega - \hat{\phi}_0)^2 \right) = \frac{1}{2\beta}\left( (\omega - \hat{\phi}_0)^2 - (\omega - \hat{\phi}_\beta)^2 \right). \end{aligned} \tag{20}$$

### S1.4   Application to the top-down modulation model

The structure of the learning and evaluation losses for the top-down modulation model is the following:

$$\begin{aligned} L^{\mathrm{learn}}(\phi, \theta) &= l^{\mathrm{learn}}(h(\phi, \theta)) \\ L^{\mathrm{eval}}(\phi, \theta) &= l^{\mathrm{eval}}(h(\phi, \theta)), \end{aligned} \tag{21}$$

where $l^{\text{learn}}(\psi)$ and $l^{\text{eval}}(\psi)$ are two data-driven losses that use learning and evaluation datasets to evaluate the performance of a network parametrized by $\psi$, and $h(\phi, \theta)$ produces the parameters $\psi$ by modulating a base network parametrized by $\theta$. Specifically, we modulate the rectified linear unit (ReLU) activation function for each neuron $i$ with a gain $g_i$ and shift $b_i$, $\sigma_\phi(x_i) = g_i((\theta \cdot x)_i - b_i)_+$, with the gain and shift parameters of all neurons defining the fast parameters $\phi = \{g, b\}$.

Applying our contrastive update (15) to this model we obtain the following equations:

$$
\begin{aligned}
\Delta\theta &= -\frac{1}{\beta}\left(\frac{\partial\mathcal{L}}{\partial\theta}(\hat{\phi}_\beta, \theta, \beta) - \frac{\partial\mathcal{L}}{\partial\theta}(\hat{\phi}_0, \theta, 0)\right)^\top \\
&= -\frac{1}{\beta}\left(\frac{\partial[l^{\text{learn}} + \beta l^{\text{eval}}]}{\partial\psi}(h(\hat{\phi}_\beta, \theta))\frac{\partial h}{\partial\theta}(\hat{\phi}_\beta, \theta) - \frac{\partial l^{\text{learn}}}{\partial\psi}(h(\hat{\phi}_0, \theta))\frac{\partial h}{\partial\theta}(\hat{\phi}_0, \theta)\right)^\top .
\end{aligned}
\tag{22}
$$

Let us now decompose what this update means. The losses $l^{\text{learn}}(\psi)$ and $[l^{\text{learn}} + \beta l^{\text{eval}}](\psi)$ measure the performance of a network parametrized by $\psi$ on the learning data, and on a weighted mix of learning and evaluation data. The derivatives $\partial_\psi l^{\text{learn}}$ and $\partial_\psi[l^{\text{learn}} + \beta l^{\text{eval}}]$ can therefore be computed using the backpropagation-of-error algorithm, or any biologically plausible alternative to it. Those derivatives are then multiplied by $\partial_\theta h$, which is a diagonal matrix as the modulation does not combine weights together, but only individually changes them. As a result, the update (22) contrasts two elementwise modulated gradients with respect to the weights.

## S2 Review of implicit gradient methods for meta-learning

The gradient associated with the bilevel optimization problem of Eq. 11 can be calculated analytically using the implicit function theorem [24]. This insight forms the basis for implicit gradient methods for meta-learning which we shortly review in the following. We additionally provide a comparison of the computational and memory complexity of different meta-learning methods in Table S1.

As for the derivation of the contrastive meta-learning rule, we start by characterizing the implicit function $\phi_\theta^*$ of $\theta$ through its corresponding first-order stationarity condition

$$\frac{\partial L^{\text{learn}}}{\partial \phi}(\phi_\theta^*, \theta) = 0. \tag{23}$$

Then, when the Hessian $\partial_\phi^2 L^{\text{learn}}(\phi_\theta^*, \theta)$ is invertible, we have

$$\begin{aligned}
\frac{\mathrm{d}}{\mathrm{d}\theta} L^{\text{eval}}(\phi_\theta^*, \theta) &= \frac{\partial L^{\text{eval}}}{\partial \theta}(\phi_\theta^*, \theta) + \frac{\partial L^{\text{eval}}}{\partial \phi} \frac{\mathrm{d}\phi_\theta^*}{\mathrm{d}\theta} \\
&= \frac{\partial L^{\text{eval}}}{\partial \theta}(\phi_\theta^*, \theta) - \frac{\partial L^{\text{eval}}}{\partial \phi} \left( \frac{\partial^2 L^{\text{learn}}}{\partial \phi^2}(\phi_\theta^*, \theta) \right)^{-1} \frac{\partial^2 L^{\text{learn}}}{\partial \phi \partial \theta}(\phi_\theta^*, \theta),
\end{aligned} \tag{24}$$

where in the first line we used the chain rule and in the second line the differentiation formula provided by the implicit function theorem [24].

In most practical applications, $\phi$ is high dimensional rendering the computation and inversion of the Hessian $\partial_\phi^2 L^{\text{learn}}(\phi_\theta^*, \theta)$ intractable. In order to obtain a practical algorithm, implicit gradient methods numerically approximate the row vector

$$\mu := -\frac{\partial L^{\text{eval}}}{\partial \phi}(\phi_\theta^*, \theta) \left( \frac{\partial^2 L^{\text{learn}}}{\partial \phi^2}(\phi_\theta^*, \theta) \right)^{-1}. \tag{25}$$

The simplest algorithm, T1-T2 [53], replaces the inverse Hessian by the identity, i.e. $\mu \approx \partial_\phi L^{\text{eval}}(\phi_\theta^*, \theta)$, which yields an estimate relying only on first derivatives.

The recurrent backpropagation algorithm [RBP, 49–51], also known as Neumann series approximation [23, 51], builds on the insight that $\mu$ is the solution of the linear system

$$x \frac{\partial^2 L^{\text{learn}}}{\partial \phi^2}(\phi_\theta^*, \theta) = -\frac{\partial L^{\text{eval}}}{\partial \phi}(\phi_\theta^*, \theta). \tag{26}$$

which can be solved via fixed point iteration.

Finally, $\mu$ can be seen as the solution of the optimization problem

$$\min_x x \frac{\partial^2 L^{\text{learn}}}{\partial \phi^2}(\phi_\theta^*, \theta) x^\top + x \frac{\partial L^{\text{eval}}}{\partial \phi}(\phi_\theta^*, \theta)^\top \tag{27}$$

when the Hessian of $L^{\text{learn}}$ is positive definite. This optimization problem can be efficiently solved via the conjugate gradient method [22, 91].

The three algorithms described above provide different estimates for $\mu$ but all follow the same basic procedure: (1) minimize the learning loss to approximate $\phi_\theta^*$; (2) estimate $\mu$; and (3) update the meta-parameters using (24) with the estimated $\mu$.

Compared to our contrastive meta-learning rule, these algorithms require a second phase that is completely different from the first one and which involves second derivatives (apart from the biased T1-T2). Additionally, as mentioned in Section 5.2, the conjugate gradient method is faster in theory, but was reported to be unstable by several studies [51, 54]. We confirm those findings in our experiments (cf. Section S3.6 and S4.1).

Table S1: Comparison of computational and memory complexity of meta-learning methods. $T$ denotes the number of steps in the base learning process and $K$ refers to steps taken in an algorithm-specific second phase. "HVP" abbreviates "Hessian-vector product" and "cross der. VP" denotes "cross derivative vector product". An algorithm is "exact in the limit" if it computes the meta-gradient or can approximate it with arbitrary precision given enough compute. The algorithms compared in this table are contrastive meta-learning (CML), conjugate gradients (CG; used in iMAML), recurrent backpropagation (RBP), T1-T2, backpropagation-through-learning (BPTL; used in MAML), its truncated version (TBPTL) and its first-order version where all Hessians are replaced by the identity (FOBPTL; also known as FOMAML) and Reptile. The first four algorithms assume that the base learning process reaches an equilibrium, whereas the last four require no such assumption. * Reptile is not a general-purpose meta-learning method as it is restricted to meta-learn the initialization of the learning process.

| Method | # gradients w.r.t. | | # 2nd-order terms | | Memory | Exact in the limit |
| | $\phi$ | $\theta$ | HVP | cross der. VP | | |
|---|---|---|---|---|---|---|
| CML (ours) | $T+K$ | 2 | 0 | 0 | $\mathcal{O}(|\phi|+|\theta|)$ | ✓ |
| CG [21, 22, 52] | $T+1$ | 1 | $K$ | 1 | $\mathcal{O}(|\phi|+|\theta|)$ | ✓ |
| RBP [23, 49–51] | $T+1$ | 1 | $K$ | 1 | $\mathcal{O}(|\phi|+|\theta|)$ | ✓ |
| T1-T2 [53] | $T+1$ | 1 | 0 | 1 | $\mathcal{O}(|\phi|+|\theta|)$ | ✗ |
| BPTL [18] | $T+1$ | $T+1$ | $T$ | 0 | $\mathcal{O}(T|\phi|+|\theta|)$ | ✓ |
| TBPTL [54] | $T+1$ | $K+1$ | $K$ | 0 | $\mathcal{O}(K|\phi|+|\theta|)$ | ✗ |
| FOBPTL [18] | $T+1$ | $T+1$ | 0 | 0 | $\mathcal{O}(T|\phi|+|\theta|)$ | ✗ |
| Reptile* [58] | $T$ | 0 | 0 | 0 | $\mathcal{O}(|\phi|)$ | ✗ |

# S3 Theoretical results

The contrastive meta-learning rule (15) only provides an approximation $\widehat{\nabla}_\theta$ to the meta-gradient $\nabla_\theta$ due to the limited precision of the fixed points and the finite difference estimator. We can in principle arbitrarily improve the approximation by spending more compute to refine the quality of the solutions $\hat{\phi}_0$ and $\hat{\phi}_\beta$ and decreasing the nudging strength $\beta$. The purpose of this section is to theoretically analyze the impact of such a refinement on the quality of the meta-gradient estimate. We state Theorem 1 formally, present a corollary of this result, and verify that it holds experimentally.

## S3.1 Meta-gradient estimation error bound

We start by upper bounding the meta-gradient estimation error $\|\widehat{\nabla}_\theta - \nabla_\theta\|$, given the value of $\beta$ and the error made in the approximation of the solutions of the lower-level learning process. Two conflicting phenomena impact the estimation error. First, our meta-learning rule uses potentially inexact solutions. Second, the finite difference approximation of the $\beta$-derivative yields the so-called finite difference error. To study those two errors in more detail, we introduce

$$\widehat{\nabla}_\theta^* := \frac{1}{\beta}\left(\frac{\partial \mathcal{L}}{\partial \theta}(\phi_{\theta,\beta}^*, \theta, \beta) - \frac{\partial \mathcal{L}}{\partial \theta}(\phi_{\theta,0}^*, \theta, 0)\right),$$

the contrastive estimate of the meta-gradient $\nabla_\theta$, but evaluated at the exact solutions $\phi_{\theta,0}^*$ and $\phi_{\theta,\beta}^*$ (recall that $\widehat{\nabla}_\theta$ has the same structure, but it is evaluated on the approximate solutions $\hat{\phi}_0$ and $\hat{\phi}_\beta$). Equipped with $\widehat{\nabla}_\theta^*$, we now have a way to quantify the two errors described above: $\|\nabla_\theta - \widehat{\nabla}_\theta^*\|$ measures the finite difference error and $\|\widehat{\nabla}_\theta^* - \widehat{\nabla}_\theta\|$ measures the solution approximation induced error, that is the consequence of the imperfect solutions.

Informally, higher $\beta$ values will reduce the sensitivity to crude approximations to the lower-level solutions while increasing the finite difference error. Theorem 1 theoretically justifies this intuition under the idealized regime of strong convexity and smoothness defined in Assumption 1. This result holds for every rule induced by equilibrium propagation.

**Assumption 1.** *Assume that $L^{\text{learn}}$ and $L^{\text{eval}}$ are three-times continuously differentiable and that they, as functions of $\phi$, verify the following properties.*

    *i. $\partial_\theta L^{\text{learn}}$ is $B^{\text{learn}}$-Lipschitz and $\partial_\theta L^{\text{eval}}$ is $B^{\text{eval}}$-Lipschitz.*

    *ii. $L^{\text{learn}}$ and $L^{\text{eval}}$ are $L$-smooth and $\mu$-strongly convex.*

    *iii. their Hessians are $\rho$-Lipschitz.*

    *iv. $\partial_\phi \partial_\theta L^{\text{learn}}$ and $\partial_\phi \partial_\theta L^{\text{eval}}$ are $\sigma$-Lipschitz.*

**Theorem 1** (Formal). *Let $\beta > 0$ and $(\delta, \delta')$ be such that*

$$\|\phi_{\theta,0}^* - \hat{\phi}_0\| \leq \delta, \quad \text{and} \quad \|\phi_{\theta,\beta}^* - \hat{\phi}_\beta\| \leq \delta'.$$

*Under Assumption 1, there exists a $\theta$-dependent constant $C$ such that*

$$\|\nabla_\theta - \widehat{\nabla}_\theta\| \leq \frac{B^{\text{learn}}(\delta + \delta')}{\beta} + B^{\text{eval}}\delta' + C\frac{\beta}{1+\beta} =: \mathcal{B}(\delta, \delta', \beta).$$

*If we additionally assume that $\theta$ lies in a compact set, we can choose $C$ to be independent of $\theta$.*

We visualize our bound in Fig. S1, as a function of $\beta$ and of the solution approximation errors $\delta$ and $\delta'$. When $\delta$ and $\delta'$ are fixed, the estimation error quickly increases when $\beta$ deviates from its optimal value and it saturates for large $\beta$ values (cf. Fig. S1A and B). A better solution approximation naturally improves the quality of the meta-gradient estimate for $\beta$ held constant (cf. Fig. S1C). However, the benefits saturate above some $\beta$-dependent value: investing extra compute in the approximation of the fixed point does not pay off if $\beta$ is not decreased accordingly.

## S3.2 Proof of Theorem 1

As mentioned above, Theorem 1 can be proved by individually bounding the two kind of errors that compose the meta-gradient estimation error, that are the finite difference error and the solution approximation induced error.

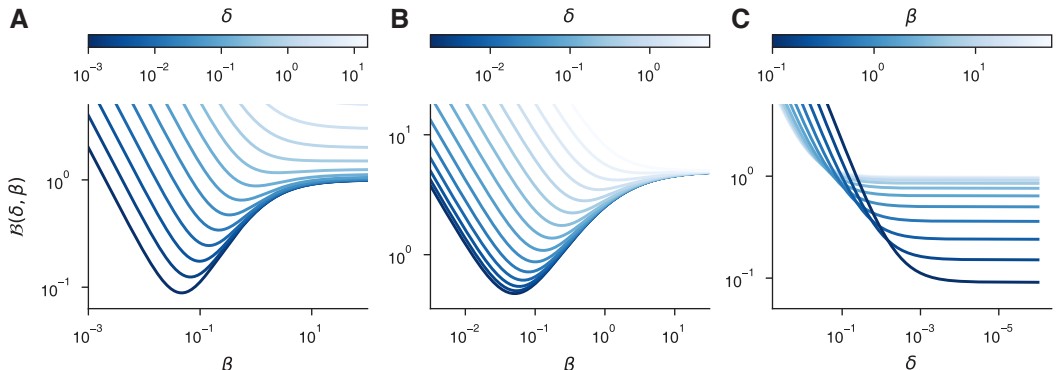

Figure S1: Theorem 1 ($C = 1$), as a function of $\beta$ (A, B) and as a function of $\delta = \delta'$ (C). (A) We take $B^{\text{learn}} = B^{\text{eval}} = 1$ and $\delta = \delta'$. (B) Bound for the setting in which $\delta'$ is fixed to 0.01 and $L^{\text{eval}}$ is independent of $\theta$ (as for the complex synapse model). (C) We use the same setting as for (A).

The $B^{\text{learn}}(\delta + \delta')/\beta + B^{\text{eval}}\delta'$ part of the bound stems from the solution approximation error, and can be obtained by using the assumption that the partial derivatives of $L^{\text{learn}}$ and $L^{\text{eval}}$ are Lipschitz continuous.

Bounding the finite difference error requires more work. We use Taylor's theorem to show that $\widehat{\nabla}_\theta^* - \nabla_\theta$ is equal to some integral remainder. It then remains to bound what is inside the integral remainder, which is the second order derivative $\mathrm{d}_\beta^2 \partial_\theta \mathcal{L}(\phi_\beta^*, \beta)$. This is done in the Lemmas presented in this section: Lemma 1 allows us to get uniform bounds, Lemmas 2 and 3 bound the first and second order derivatives of $\beta \mapsto \phi_\beta^*$ and Lemma 4 bounds $\mathrm{d}_\beta^2 \partial_\theta \mathcal{L}(\phi_\beta^*, \beta)$ with the norm of the two derivatives we have just bounded. We present the proofs for those four lemmas in Section S3.3.

**Lemma 1.** *Under Assumption 1.ii, if $\theta$ lies in a compact set $\mathcal{D}$ the function $(\theta, \beta) \mapsto \phi_{\theta,\beta}^*$ is uniformly bounded.*

**Lemma 2.** *Under Assumption 1.ii, there exists a $\theta$-dependent constant $R$ s.t., for every positive $\beta$,*

$$\left\| \frac{\mathrm{d}\phi_\beta^*}{\mathrm{d}\beta} \right\| \leq \frac{LR}{(1+\beta)^2 \mu}.$$

*If we additionally assume that $\theta$ lies in a compact set, we can choose $R$ to be independent of $\theta$.*

**Remark 1.** *A side product of the proof of Lemma 2 is a bound on the distance between the minimizer of $\mathcal{L}$ and the minimizers of $L^{\text{learn}}$ and $L^{\text{eval}}$. We have*

$$\|\phi_\beta^* - \phi_\infty^*\| \leq \frac{1}{1+\beta}$$

*and*

$$\|\phi_\beta^* - \phi_0^*\| \leq \frac{\beta}{1+\beta}$$

*up to some constant factors.*

**Lemma 3.** *Under Assumptions 1.ii and 1.iii,*

$$\left\| \frac{\mathrm{d}^2\phi_\beta^*}{\mathrm{d}\beta^2} \right\| \leq \frac{\rho}{\mu} \left\| \frac{\mathrm{d}\phi_\beta^*}{\mathrm{d}\beta} \right\|^2 + \frac{2L}{(1+\beta)\mu} \left\| \frac{\mathrm{d}\phi_\beta^*}{\mathrm{d}\beta} \right\|.$$

When Lemma 3 is combined with Lemma 2,

$$\left\| \frac{\mathrm{d}^2\phi_\beta^*}{\mathrm{d}\beta^2} \right\| \leq \frac{1}{(1+\beta)^3}. \tag{28}$$

up to some constant factor.

**Lemma 4.** *Under Assumptions 1.ii, 1.iii and 1.iv, there exists a constant $M$ such that*

$$\left\| \frac{\mathrm{d}^2}{\mathrm{d}\beta^2} \frac{\partial \mathcal{L}}{\partial \theta}(\phi_\beta^*, \beta) \right\| \leq M \left( \left\| \frac{\mathrm{d}\phi_\beta^*}{\mathrm{d}\beta} \right\| + (1 + \beta) \left( \left\| \frac{\mathrm{d}\phi_\beta^*}{\mathrm{d}\beta} \right\|^2 + \left\| \frac{\mathrm{d}^2\phi_\beta^*}{\mathrm{d}\beta^2} \right\| \right) \right).$$

We can now prove Theorem 1 using the four lemmas that we have just presented. Note that we omit the $\theta$-dependency whenever $\theta$ is fixed, for the sake of conciseness.

*Proof of Theorem 1.* We separate the sources of error within the meta-gradient estimation error using the triangle inequality:

$$\|\widehat{\nabla}_\theta - \nabla_\theta\| \leq \underbrace{\|\widehat{\nabla}_\theta - \widehat{\nabla}_\theta^*\|}_{a)} + \underbrace{\|\widehat{\nabla}_\theta^* - \nabla_\theta\|}_{b)}, \tag{29}$$

and bound the two terms separately:

a) Recall that

$$\widehat{\nabla}_\theta = \frac{1}{\beta} \left( \frac{\partial \mathcal{L}}{\partial \theta}(\hat{\phi}_\beta, \beta) - \frac{\partial \mathcal{L}}{\partial \theta}(\hat{\phi}_0, 0) \right) \tag{30}$$

and that a similar formula holds for $\widehat{\nabla}_\theta^*$ (evaluated at the true solutions instead of the approximations). It follows

$$\|\widehat{\nabla}_\theta - \widehat{\nabla}_\theta^*\| \leq \frac{1}{\beta} \left( \left\| \frac{\partial \mathcal{L}}{\partial \theta}(\hat{\phi}_\beta, \beta) - \frac{\partial \mathcal{L}}{\partial \theta}(\phi_\beta^*, \beta) \right\| + \left\| \frac{\partial \mathcal{L}}{\partial \theta}(\hat{\phi}_0, 0) - \frac{\partial \mathcal{L}}{\partial \theta}(\phi_0^*, 0) \right\| \right). \tag{31}$$

Since $\phi \mapsto \partial_\theta \mathcal{L}(\phi, \beta)$ is a $(B^{\mathrm{learn}} + \beta B^{\mathrm{eval}})$-Lipschitz function as a sum of $\partial_\theta L^{\mathrm{learn}}$ and $\partial_\theta L^{\mathrm{eval}}$, two Lipschitz continuous functions with constants $B^{\mathrm{learn}}$ and $B^{\mathrm{eval}}$,

$$\|\widehat{\nabla}_\theta - \widehat{\nabla}_\theta^*\| \leq \frac{B^{\mathrm{learn}} + \beta B^{\mathrm{eval}}}{\beta} \|\hat{\phi}_\beta - \phi_\beta^*\| + \frac{B^{\mathrm{learn}}}{\beta} \|\hat{\phi}_0 - \phi_0^*\| \tag{32}$$

$$\leq \frac{B^{\mathrm{learn}} + \beta B^{\mathrm{eval}}}{\beta} \delta' + \frac{B^{\mathrm{learn}}}{\beta} \delta. \tag{33}$$

b) Taylor's theorem applied to $\beta \mapsto \partial_\theta \mathcal{L}(\phi_\beta^*, \beta)$ up to the first order of differentiation yields

$$\frac{\partial \mathcal{L}}{\partial \theta}(\phi_\beta^*, \beta) = \frac{\partial \mathcal{L}}{\partial \theta}(\phi_0^*, 0) + \beta \frac{\mathrm{d}}{\mathrm{d}\beta} \frac{\partial \mathcal{L}}{\partial \theta}(\phi_0^*, 0) + \int_0^\beta (\beta - t) \frac{\mathrm{d}^2}{\mathrm{d}\beta^2} \frac{\partial \mathcal{L}}{\partial \theta}(\phi_t^*, t) \, \mathrm{d}t. \tag{34}$$

The equilibrium propagation theorem (Theorem S1), which is applicable thanks to Assumption 1.ii, gives

$$\nabla_\theta = \frac{\mathrm{d}}{\mathrm{d}\beta} \frac{\partial \mathcal{L}}{\partial \theta}(\phi_0^*, 0), \tag{35}$$

hence

$$\|\widehat{\nabla}_\theta^* - \nabla_\theta\| = \left\| \int_0^\beta (\beta - t) \frac{\mathrm{d}^2}{\mathrm{d}\beta^2} \frac{\partial \mathcal{L}}{\partial \theta}(\phi_t^*, t) \, \mathrm{d}t \right\|. \tag{36}$$

Using the integral version of the Cauchy-Schwartz inequality, we have

$$\|\widehat{\nabla}_\theta^* - \nabla_\theta\| \leq \int_0^\beta (\beta - t) \left\| \frac{\mathrm{d}^2}{\mathrm{d}\beta^2} \frac{\partial \mathcal{L}}{\partial \theta}(\phi_t^*, t) \right\| \mathrm{d}t. \tag{37}$$

We now use Lemma 4 combined with Lemmas 2 and 3 to bound $\mathrm{d}_\beta^2 \partial_\theta \mathcal{L}(\phi_t^*, t)$. We focus on the $\beta$ dependencies and omitting constant factors:

$$\left\| \frac{\mathrm{d}^2}{\mathrm{d}\beta^2} \frac{\partial \mathcal{L}}{\partial \theta}(\phi_t^*, t) \right\| \leq \left\| \frac{\mathrm{d}\phi_t^*}{\mathrm{d}\beta} \right\| + (1 + t) \left( \left\| \frac{\mathrm{d}\phi_t^*}{\mathrm{d}\beta} \right\|^2 + \left\| \frac{\mathrm{d}^2\phi_t^*}{\mathrm{d}\beta^2} \right\| \right)$$

$$\leq \frac{1}{(1 + t)^2} + (1 + t) \left( \frac{1}{(1 + t)^3} + \frac{1}{(1 + t)^4} \right)$$

$$\leq (1 + t)^{-2}.$$

It follows that

$$
\begin{aligned}
\|\widehat{\nabla}_\theta^* - \nabla_\theta\| &\leq \int_0^\beta \frac{(\beta - t)}{(1+t)^2} dt \\
&= (1+\beta) \int_0^\beta \frac{1}{(1+t)^2} dt - \int_0^\beta \frac{1}{(1+t)} dt \\
&= (1+\beta) \frac{\beta}{1+\beta} - \ln(1+\beta) \\
&\leq \beta - \frac{\beta}{1+\beta} \\
&= \frac{\beta^2}{1+\beta}.
\end{aligned}
\tag{38}
$$

where the inequality comes from the well-known $\ln(x) \geq 1 - \frac{1}{x}$ inequality for positive $x$ (applied to $x = 1 + \beta$). There hence exists a constant $C$ such that

$$
\|\widehat{\nabla}_\theta^* - \nabla_\theta\| \leq C \frac{\beta}{1+\beta}.
\tag{39}
$$

If $\theta$ lies in a compact set, the bound in Lemma 2 is uniform over $\theta$. This is the only constant factor that depends on $\theta$, so the bound is uniform. $\qquad\square$

### S3.3  Proof of technical lemmas

In this section, we prove the four technical lemmas that we need for Theorem 1.

**Proof of Lemma 1**

**Lemma 1.** *Under Assumption 1.ii, if $\theta$ lies in a compact set $\mathcal{D}$ the function $(\theta, \beta) \mapsto \phi_{\theta,\beta}^*$ is uniformly bounded.*

*Proof.* Let $\alpha \in [0, 1]$. Define

$$
\mathcal{L}'(\phi, \theta, \alpha) := (1-\alpha) L^{\text{learn}}(\phi, \theta) + \alpha L^{\text{eval}}(\phi, \theta).
\tag{40}
$$

As $L^{\text{learn}}$ and $L^{\text{eval}}$ are strongly-convex, there exists a unique minimizer $\phi_{\theta,\alpha}^{*\prime}$ of $\phi \mapsto \mathcal{L}'(\phi, \theta, \alpha)$. The implicit function theorem ensures that the function $(\theta, \alpha) \mapsto \phi_{\theta,\alpha}^{*\prime}$, defined on $\mathcal{D} \times [0, 1]$, is continuous. As $\mathcal{D} \times [0, 1]$ is a compact set, $\phi_{\theta,\alpha}^{*\prime}$ is then uniformly bounded. Now, remark that

$$
\mathcal{L}(\phi, \theta, \beta) = (1+\beta) \mathcal{L}'\left(\phi, \theta, \frac{\beta}{1+\beta}\right)
\tag{41}
$$

and thus $\phi_{\theta,\beta}^* = \phi_{\theta,\beta/(1+\beta)}^{*\prime}$. It follows that $\phi_{\theta,\beta}^*$ is uniformly bounded. $\qquad\square$

**Proof of Lemma 2**

**Lemma 2.** *Under Assumption 1.ii, there exists a $\theta$-dependent constant $R$ s.t., for every positive $\beta$,*

$$
\left\| \frac{d\phi_\beta^*}{d\beta} \right\| \leq \frac{LR}{(1+\beta)^2 \mu}.
$$

*If we additionally assume that $\theta$ lies in a compact set, we can choose $R$ to be independent of $\theta$.*

*Proof.* The function $\phi \mapsto \mathcal{L}(\phi, \beta)$ is $(1+\beta)\mu$-strongly convex so its Hessian $\partial_\phi^2 \mathcal{L}$ is invertible and its inverse has a spectral norm upper bounded by $1/((1+\beta)\mu)$. The use of the implicit function theorem follows and gives

$$
\begin{aligned}
\left\| d_\beta \phi_\beta^* \right\| &= \left\| -\left( \partial_\phi^2 \mathcal{L}(\phi_\beta^*, \beta) \right)^{-1} \partial_\beta \partial_\phi \mathcal{L}(\phi_\beta^*) \right\| \\
&= \left\| -\left( \partial_\phi^2 \mathcal{L}(\phi_\beta^*, \beta) \right)^{-1} \partial_\phi L^{\text{eval}}(\phi_\beta^*) \right\| \\
&\leq \frac{1}{(1+\beta)\mu} \left\| \partial_\phi L^{\text{eval}}(\phi_\beta^*) \right\|.
\end{aligned}
\tag{42}
$$

It remains to bound the gradient of $L^{\text{eval}}$. Since $\beta \mapsto \phi^*_\beta$ is continuous and has finite limits in 0 and $\infty$ (namely the minimizers of $L^{\text{learn}}$ and $L^{\text{eval}}$), it evolves in a bounded set. There hence exists a positive constant $R$ such that, for all positive $\beta$,

$$\max\left(\left\|\phi^*_\beta - \phi^*_0\right\|, \left\|\phi^*_\beta - \phi^*_\infty\right\|\right) \leq \frac{R}{2}. \tag{43}$$

If $\theta$ lies in a compact set, Lemma 1 guarantees that there exists such a constant that doesn't depend on the choice of $\theta$. We then bound the gradient of $L^{\text{eval}}$ using the smoothness properties of $L^{\text{learn}}$ and $L^{\text{eval}}$, either directly

$$\|\partial_\phi L^{\text{eval}}(\phi^*_\beta)\| \leq L\|\phi^*_\beta - \phi^*_\infty\| \leq \frac{LR}{2} \tag{44}$$

or indirectly, using the fixed point condition $\partial_\phi \mathcal{L}(\phi^*_\beta, \beta) = 0$,

$$\|\partial_\phi L^{\text{eval}}(\phi^*_\beta)\| = \frac{1}{\beta}\|-\partial_\phi L^{\text{learn}}(\phi^*_\beta)\| \leq \frac{L\|\phi^*_\beta - \phi^*_0\|}{\beta} \leq \frac{LR}{2\beta}. \tag{45}$$

The required result is finally obtained by remarking

$$\|\partial_\phi L^{\text{eval}}(\phi^*_\beta)\| \leq \min\left(1, \frac{1}{\beta}\right)\frac{LR}{2} \leq \frac{LR}{1+\beta}. \tag{46}$$

$\square$

**Proof of Remark 1**   We now prove Remark 1, which directly follows from the previous proof. Recall that we have just proved

$$\|\partial_\phi L^{\text{eval}}(\phi^*_\beta)\| \leq \frac{LR}{1+\beta}. \tag{47}$$

With the strong convexity of $L^{\text{eval}}$, the gradient is also lower bounded

$$\|\partial_\phi L^{\text{eval}}(\phi^*_\beta)\| \geq \mu\|\phi^*_\beta - \phi^*_\infty\|, \tag{48}$$

meaning that

$$\|\phi^*_\beta - \phi^*_\infty\| \leq \frac{LR}{\mu(1+\beta)}. \tag{49}$$

Similarly, one can show that

$$\|\phi^*_0 - \phi^*_\beta\| \leq \frac{\beta}{1+\beta} \tag{50}$$

up to some constant factor. This can be proved with

$$\|\phi^*_0 - \phi^*_\beta\| \leq \frac{\|\partial_\phi L^{\text{learn}}(\phi^*_\beta)\|}{\mu} = \frac{\beta\|\partial_\phi L^{\text{eval}}(\phi^*_\beta)\|}{\mu} \leq \frac{\beta LR}{(1+\beta)\mu}. \tag{51}$$

**Proof of Lemma 3**

**Lemma 3.** *Under Assumptions 1.ii and 1.iii,*

$$\left\|\frac{\mathrm{d}^2\phi^*_\beta}{\mathrm{d}\beta^2}\right\| \leq \frac{\rho}{\mu}\left\|\frac{\mathrm{d}\phi^*_\beta}{\mathrm{d}\beta}\right\|^2 + \frac{2L}{(1+\beta)\mu}\left\|\frac{\mathrm{d}\phi^*_\beta}{\mathrm{d}\beta}\right\|.$$

*Proof.* The starting point of the proof is the implicit function theorem, that we differentiate with respect to $\beta$ as a product of functions

$$\begin{aligned}
\frac{\mathrm{d}^2\phi^*_\beta}{\mathrm{d}\beta^2} &= \frac{\mathrm{d}}{\mathrm{d}\beta}\left(-\left(\frac{\partial^2\mathcal{L}}{\partial\phi^2}(\phi^*_\beta, \beta)\right)^{-1}\frac{\partial L^{\text{eval}}}{\partial\phi}(\phi^*_\beta)\right) \\
&= -\underbrace{\left(\frac{\mathrm{d}}{\mathrm{d}\beta}\frac{\partial^2\mathcal{L}}{\partial\phi^2}(\phi^*_\beta, \beta)^{-1}\right)\frac{\partial L^{\text{eval}}}{\partial\phi}(\phi^*_\beta)}_{a)} - \underbrace{\frac{\partial^2\mathcal{L}}{\partial\phi^2}(\phi^*_\beta, \beta)^{-1}\left(\frac{\mathrm{d}}{\mathrm{d}\beta}\frac{\partial L^{\text{eval}}}{\partial\phi}(\phi^*_\beta)\right)}_{b)}.
\end{aligned} \tag{52}$$

We now individually calculate and bound each term.

a) The differentiation of the inverse of a matrix gives

$$a) = -\partial_\phi^2 \mathcal{L}(\phi_\beta^*, \beta)^{-1} \left( \mathrm{d}_\beta \partial_\phi^2 \mathcal{L}(\phi_\beta^*, \beta) \right) \partial_\phi^2 \mathcal{L}(\phi_\beta^*, \beta)^{-1} \partial_\phi L^{\mathrm{eval}}(\phi_\beta^*), \tag{53}$$

which we can rewrite as

$$a) = \partial_\phi^2 \mathcal{L}(\phi_\beta^*, \beta)^{-1} \left( \mathrm{d}_\beta \partial_\phi^2 \mathcal{L}(\phi_\beta^*, \beta) \right) \mathrm{d}_\beta \phi_\beta^*. \tag{54}$$

The derivative term in the middle of the right hand side is equal to

$$\begin{aligned}
\mathrm{d}_\beta \partial_\phi^2 \mathcal{L}(\phi_\beta^*, \beta) &= \mathrm{d}_\beta \left[ \partial_\phi^2 L^{\mathrm{learn}}(\phi_\beta^*) + \beta \partial_\phi^2 L^{\mathrm{eval}}(\phi_\beta^*) \right] \\
&= \mathrm{d}_\beta \partial_\phi^2 L^{\mathrm{learn}}(\phi_\beta^*) + \beta \mathrm{d}_\beta \partial_\phi^2 L^{\mathrm{eval}}(\phi_\beta^*) + \partial_\phi^2 L^{\mathrm{eval}}(\phi_\beta^*).
\end{aligned} \tag{55}$$

Using the Lipschitz continuity of the Hessians,

$$\left\| \mathrm{d}_\beta \partial_\phi^2 L^{\mathrm{learn}}(\phi_\beta^*) + \beta \mathrm{d}_\beta \partial_\phi^2 L^{\mathrm{eval}}(\phi_\beta^*) \right\| \leq (1 + \beta)\rho \left\| \mathrm{d}_\beta \phi_\beta^* \right\|. \tag{56}$$

We can upper bound the norm of the Hessian of $L^{\mathrm{eval}}$ by $L$ as $L^{\mathrm{eval}}$ is L-smooth. The last two equations hence give

$$\left\| \mathrm{d}_\beta \partial_\phi^2 \mathcal{L}(\phi_\beta^*, \beta) \right\| \leq (1 + \beta)\rho \left\| \mathrm{d}_\beta \phi_\beta^* \right\| + L. \tag{57}$$

We finally have

$$\begin{aligned}
\|a)\| &\leq \frac{1}{\mu(1 + \beta)} \left( (1 + \beta)\rho \left\| \mathrm{d}_\beta \phi_\beta^* \right\| + L \right) \left\| \mathrm{d}_\beta \phi_\beta^* \right\| \\
&\leq \frac{\rho}{\mu} \left\| \mathrm{d}_\beta \phi_\beta^* \right\|^2 + \frac{L}{(1 + \beta)\mu} \left\| \mathrm{d}_\beta \phi_\beta^* \right\|.
\end{aligned} \tag{58}$$

b) With the chain rule,

$$\mathrm{d}_\beta \partial_\phi L^{\mathrm{eval}}(\phi_\beta^*) = \partial_\phi^2 L^{\mathrm{eval}}(\phi_\beta^*) \mathrm{d}_\beta \phi_\beta^* \tag{59}$$

so

$$\begin{aligned}
\|b)\| &\leq \left\| \partial_\phi^2 \mathcal{L}(\phi_\beta^*, \beta)^{-1} \right\| \left\| \partial_\phi^2 L^{\mathrm{eval}}(\phi_\beta^*) \right\| \left\| \mathrm{d}_\beta \phi_\beta^* \right\| \\
&\leq \frac{L}{(1 + \beta)\mu} \left\| \mathrm{d}_\beta \phi_\beta^* \right\|.
\end{aligned} \tag{60}$$

$\square$

**Proof of Lemma 4**

**Lemma 4.** *Under Assumptions 1.ii, 1.iii and 1.iv, there exists a constant $M$ such that*

$$\left\| \frac{\mathrm{d}^2}{\mathrm{d}\beta^2} \frac{\partial \mathcal{L}}{\partial \theta}(\phi_\beta^*, \beta) \right\| \leq M \left( \left\| \frac{\mathrm{d}\phi_\beta^*}{\mathrm{d}\beta} \right\| + (1 + \beta) \left( \left\| \frac{\mathrm{d}\phi_\beta^*}{\mathrm{d}\beta} \right\|^2 + \left\| \frac{\mathrm{d}^2 \phi_\beta^*}{\mathrm{d}\beta^2} \right\| \right) \right).$$

*Proof.* We want to bound the norm of $\mathrm{d}_\beta^2 \partial_\theta \mathcal{L}(\phi_\beta^*, \beta)$. The first order derivative can be calculated with the chain rule of differentiation

$$\mathrm{d}_\beta \partial_\theta \mathcal{L}(\phi_\beta^*, \beta) = \partial_\beta \partial_\theta \mathcal{L}(\phi_\beta^*, \beta) + \partial_\phi \partial_\theta \mathcal{L}(\phi_\beta^*, \beta) \mathrm{d}_\beta \phi_\beta^*. \tag{61}$$

We then once again differentiate this equation with respect to $\beta$. The $\partial_\beta \partial_\theta \mathcal{L}(\phi_\beta^*, \beta)$ term has in fact, due to the nature of $\mathcal{L}$, no direct dependence on $\beta$ and is equal to $\partial_\theta L^{\mathrm{eval}}(\phi_\beta^*)$. Hence

$$\mathrm{d}_\beta \partial_\beta \partial_\theta \mathcal{L}(\phi_\beta^*, \beta) = \partial_\phi \partial_\theta L^{\mathrm{eval}}(\phi_\beta^*) \mathrm{d}_\beta \phi_\beta^*. \tag{62}$$

Differentiating the other term yields

$$\mathrm{d}_\beta \left[ \partial_\phi \partial_\theta \mathcal{L}(\phi_\beta^*, \beta) \mathrm{d}_\beta \phi_\beta^* \right] = \left[ \partial_\beta \partial_\phi \partial_\theta \mathcal{L}(\phi_\beta^*, \beta) + \partial_\phi^2 \partial_\theta \mathcal{L}(\phi_\beta^*, \beta) \otimes \mathrm{d}_\beta \phi_\beta^* \right] \mathrm{d}_\beta \phi_\beta^* + \\ \partial_\phi \partial_\theta \mathcal{L}(\phi_\beta^*, \beta) \mathrm{d}_\beta^2 \phi_\beta^*. \tag{63}$$

Therefore,

$$\mathrm{d}_\beta^2 \partial_\theta \mathcal{L}(\phi_\beta^*, \beta) = 2 \partial_\phi \partial_\theta L^{\mathrm{eval}}(\phi_\beta^*) \mathrm{d}_\beta \phi_\beta^* + \partial_\phi^2 \partial_\theta \mathcal{L}(\phi_\beta^*, \beta) \otimes \mathrm{d}_\beta \phi_\beta^* \otimes \mathrm{d}_\beta \phi_\beta^* + \\ \partial_\phi \partial_\theta \mathcal{L}(\phi_\beta^*, \beta) \mathrm{d}_\beta^2 \phi_\beta^*. \tag{64}$$

We now individually bound each term:

- due to Assumption 1.$i$, $\phi \mapsto \partial_\theta L^{\text{eval}}(\phi)$ is $B^{\text{eval}}$-Lipschitz continuous, so $\|\partial_\phi \partial_\theta L^{\text{eval}}\| \leq B^{\text{eval}}$ and

$$\left\|2\partial_\phi \partial_\theta L^{\text{eval}}(\phi_\beta^*)\mathrm{d}_\beta \phi_\beta^*\right\| \leq 2B^{\text{eval}} \left\|\mathrm{d}_\beta \phi_\beta^*\right\|. \tag{65}$$

- similarly to the previous point,

$$\left\|\partial_\phi \partial_\theta \mathcal{L}(\phi_\beta^*)\mathrm{d}_\beta^2 \phi_\beta^*\right\| \leq (B^{\text{learn}} + \beta B^{\text{eval}}) \left\|\mathrm{d}_\beta^2 \phi_\beta^*\right\|. \tag{66}$$

- Assumption 1.$iv$ ensures that $\phi \mapsto \partial_\phi \partial_\theta \mathcal{L}(\phi, \beta)$ is $(1+\beta)\sigma$-Lipschitz continous and

$$\left\|\partial_\phi^2 \partial_\theta \mathcal{L}(\phi_\beta^*, \beta) \otimes \mathrm{d}_\beta \phi_\beta^* \otimes \mathrm{d}_\beta \phi_\beta^*\right\| \leq (1+\beta)\sigma \left\|\mathrm{d}_\beta \phi_\beta^*\right\|^2. \tag{67}$$

Take $M := \max(2B^{\text{eval}}, B^{\text{learn}}, \sigma)$: we now have the desired result. $\qquad \square$

### S3.4 A corollary of Theorem 1

Theorem 1 highlights the importance of considering $\beta$ as a hyperparameter of the learning rule that needs to be adjusted to yield the best possible meta-gradient estimate. Corollary 1 removes the dependence in $\beta$ and considers the best achievable bound under given fixed point approximation errors.

**Corollary 1.** *Under Assumption 1, if we suppose that for every strictly positive $\beta$ we approximate the two fixed points with precision $\delta$ and $\delta'$ and if $(\delta + \delta') < C/B^{\text{learn}}$, the best achievable bound in Theorem 1 is smaller than*

$$B^{\text{eval}}\delta' + 2\sqrt{CB^{\text{learn}}(\delta + \delta')}$$

*and is attained for $\beta$ equal to*

$$\beta^*(\delta, \delta') = \frac{\sqrt{B^{\text{learn}}(\delta + \delta')}}{\sqrt{C} - \sqrt{B^{\text{learn}}(\delta + \delta')}}.$$

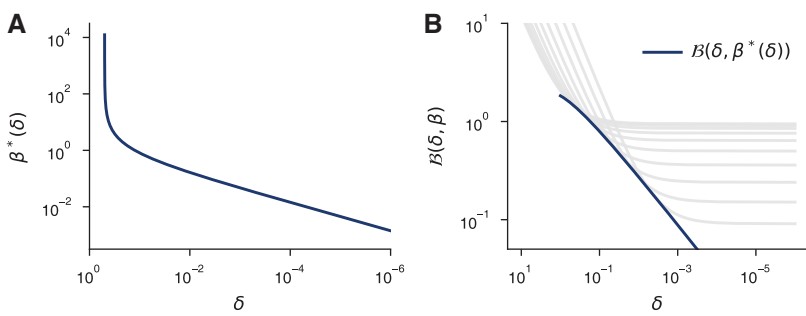

Figure S2: Visualization of Corollary 1. (A) $\beta$ value that minimizes the bound, as a function of $\delta = \delta'$. (B) Best achievable bound as a function of $\delta = \delta'$ in blue (more precisely the one before the last upper bound in the proof). The grey lines are the bounds from Theorem 1 we laid out on Fig. S1C.

The most limiting part of the bound depends on the sum $\delta + \delta'$ and not on the individual quantities, suggesting that the two errors should be of the same magnitude to avoid unnecessary computations.

*Proof.* The $\beta$ derivative of the bound $\mathcal{B}$ obtained in Theorem 1 is

$$\frac{\partial \mathcal{B}}{\partial \beta}(\delta, \delta', \beta) = -\frac{B^{\text{learn}}(\delta + \delta')}{\beta^2} + \frac{C}{(1+\beta)^2} \tag{68}$$

and vanishes for $\beta$ verifying

$$\beta\left(\sqrt{C} - \sqrt{B^{\text{learn}}(\delta + \delta')}\right) = \sqrt{B^{\text{learn}}(\delta + \delta')}. \tag{69}$$

As $(\delta + \delta') < C/B^{\text{learn}}$, the previous criterion is met when $\beta$ is equal to the positive

$$\beta^* := \frac{\sqrt{B^{\text{learn}}(\delta + \delta')}}{\sqrt{C} - \sqrt{B^{\text{learn}}(\delta + \delta')}}. \tag{70}$$

The optimal bound is then

$$\mathcal{B}(\delta, \delta', \beta^*) = B^{\text{eval}}\delta' + \sqrt{B^{\text{learn}}(\delta + \delta')}\left(\sqrt{C} - \sqrt{B^{\text{learn}}(\delta + \delta')}\right) + \sqrt{CB^{\text{learn}}(\delta + \delta')}$$

$$\leq B^{\text{eval}}\delta' + 2\sqrt{CB^{\text{learn}}(\delta + \delta')}.$$

$\square$

### S3.5 Verification of the theoretical results on an analytical problem

We investigate a quadratic approximation of the complex synapse model, in which everything can be calculated in closed form and where the assumptions needed for the theory hold. Define $L^{\text{learn}}$ and $L^{\text{eval}}$ as follows[3]:

$$L^{\text{learn}}(\phi, \omega) = \frac{1}{2}(\phi - \phi^l)^\top H(\phi - \phi^l) + \frac{\lambda}{2}\|\phi - \omega\|^2$$

$$L^{\text{eval}}(\phi) = \frac{1}{2}(\phi - \phi^e)^\top H(\phi - \phi^e)$$

where $\lambda$ is a scalar that controls the strength of the regularization that we consider fixed, $\phi^l$ and $\phi^e$ two vectors and $H$ a positive definite diagonal matrix. The rationale behind this approximation is the following: the data-driven learning and evaluation losses share the same curvature but have different minimizers, respectively $\phi^l$ and $\phi^e$. The matrix $H$ then models the Hessian and we consider it diagonal for simplicity. Thanks to the quadratic approximation, many quantities involved in our contrastive meta-learning rule can be calculated in closed form.

**Calculation of the finite difference error.** A formula for the minimizer of $\mathcal{L} = L^{\text{learn}} + \beta L^{\text{eval}}$ can be derived analytically. The derivative of $\mathcal{L}$ vanishes if and only if

$$((1 + \beta)H + \lambda\text{Id})\phi - H\phi^l - \beta H\phi^e - \lambda\omega = 0,$$

hence

$$\phi^*_{\theta,\beta} = ((1 + \beta)\text{Id} + \lambda H^{-1})^{-1}(\phi^l + \beta\phi^e + \lambda H^{-1}\omega).$$

$\lambda H^{-1}$ is an interesting quantity in this example. It acts as the effective per-coordinate regularization strength: regularization will be stronger on flat directions.

The meta-gradient calculation follows. As

$$\partial_\omega \mathcal{L}(\phi, \omega, \beta) = -\lambda(\phi - \omega),$$

the use of the equilibrium propagation theorem (Theorem S1) gives

$$\nabla_\omega = \frac{\mathrm{d}}{\mathrm{d}\beta}\frac{\partial \mathcal{L}}{\partial \omega}(\phi^*_{\theta,\beta}, \omega, \beta)\bigg|_{\beta=0}$$

$$= \frac{\partial^2 L^{\text{learn}}}{\partial\phi\partial\omega}(\phi^*_{\theta,0}, \omega)\frac{\mathrm{d}\phi^*_{\theta,\beta}}{\mathrm{d}\beta}\bigg|_{\beta=0} + 0$$

$$= -\lambda \frac{\mathrm{d}\phi^*_{\theta,\beta}}{\mathrm{d}\beta}\bigg|_{\beta=0}.$$

---

[3]In our experiments, we take the dimension of the parameter space $N$ to be equal to 50. The Hessian is taken to be $\text{diag}(1, ..., 1/N)$. $\omega$ is randomly generated according to $\omega \sim \mathcal{N}(0, \sigma_\omega)$ with $\sigma_\omega = 2$. $\phi^l$ and $\phi^e$ are drawn around $\phi^\tau \sim \mathcal{N}(0, \sigma_\tau)$ (with $\sigma_\tau = 1$).

It now remains to calculate the derivative of $\phi_{\theta,\beta}^*$ with respect to $\beta$ using the formula of $\phi_{\theta,\beta}^*$:

$$
\begin{aligned}
\frac{\mathrm{d}\phi_{\theta,\beta}^*}{\mathrm{d}\beta} &= \left((1+\beta)\mathrm{Id} + \lambda H^{-1}\right)^{-1}\phi^e \\
&\quad - \left((1+\beta)\mathrm{Id} + \lambda H^{-1}\right)^{-2}\left(\phi^l + \beta\phi^e + \lambda H^{-1}\omega\right) \\
&= \left((1+\beta)\mathrm{Id} + \lambda H^{-1}\right)^{-2}\left((1+\beta)\phi^e + \lambda H^{-1}\phi^e - \phi^l - \beta\phi^e - \lambda H^{-1}\omega\right) \\
&= \left((1+\beta)\mathrm{Id} + \lambda H^{-1}\right)^{-2}\left((\phi^e - \phi^l) + \lambda H^{-1}(\phi^e - \omega)\right)
\end{aligned}
$$

Define $\psi := (\phi^e - \phi^l) + \lambda H^{-1}(\phi^e - \omega)$; the meta-gradient finally is

$$
\nabla_\omega = -\lambda(\mathrm{Id} + \lambda H^{-1})^{-2}\psi.
$$

We can now calculate the finite difference error. Recall the equilibrium propagation estimate at fixed points

$$
\widehat{\nabla}_\omega^* = \frac{1}{\beta}\left(\frac{\partial\mathcal{L}}{\partial\omega}(\phi_{\theta,\beta}^*, \omega, \beta) - \frac{\partial\mathcal{L}}{\partial\omega}(\phi_{\theta,0}^*, \omega, 0)\right).
$$

In this formulation, it is equal to

$$
\begin{aligned}
\widehat{\nabla}_\omega^* &= -\frac{\lambda}{\beta}(\phi_{\theta,\beta}^* - \phi_{\theta,0}^*) \\
&= -\lambda\left((\mathrm{Id} + \lambda H^{-1})((1+\beta)\mathrm{Id} + \lambda H^{-1})\right)^{-1}\psi \\
&= (\mathrm{Id} + \lambda H^{-1})\left((1+\beta)\mathrm{Id} + \lambda H^{-1}\right)^{-1}\nabla_\omega.
\end{aligned}
$$

The finite difference can now be lower and upper bounded. First,

$$
\nabla_\omega - \widehat{\nabla}_\omega^* = \beta\left((1+\beta)\mathrm{Id} + \lambda H^{-1}\right)^{-1}\nabla_\omega.
$$

Introduce $\mu$ the smallest eigenvalue of $H$ and $L$ its largest one. We then have

$$
\frac{\mu\beta}{(1+\beta)\mu + \lambda}\|\nabla_\omega\| \le \|\nabla_\omega - \widehat{\nabla}_\omega^*\| \le \frac{L\beta}{(1+\beta)L + \lambda}\|\nabla_\omega\|. \tag{71}
$$

This shows that the finite difference error part of Theorem 1 is tight and, in this case, accurately describes the behavior of the finite difference error as a function of $\beta$.

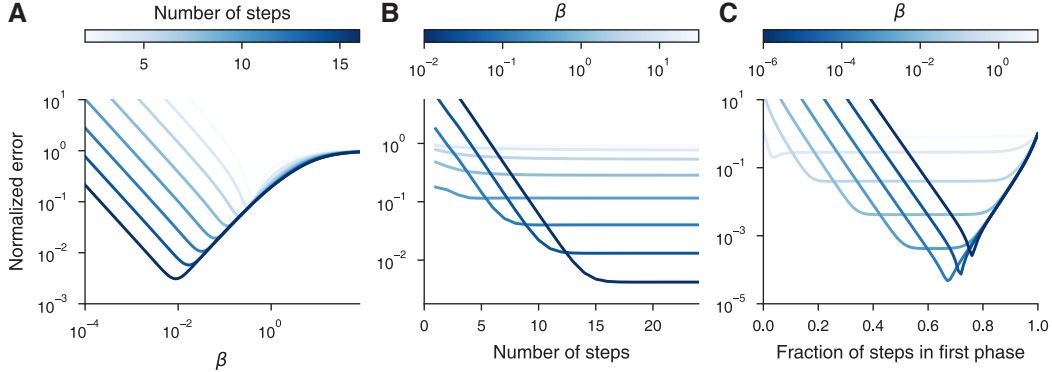

Figure S3: Empirical verification of the theoretical results on an analytical quadratic approximation of the synaptic model. We plot the normalized error between the meta-gradient estimate $\widehat{\nabla}_\theta$ and the true one $\nabla_\theta$, as a function of $\beta$ (A), of the number of steps in the two phases (which is a proxy for $-\log\delta$ and $-\log\delta'$ used in the theory) (B), and as a function of the allocation of the computational resources between the two phases, the total number of steps being fixed to 100 (C).

**Empirical results.** The solution approximation induced error part of the bound cannot be treated analytically as it depends on $\delta$ and $\delta'$, which are in essence empirical quantities. We cannot directly

control them either. Instead, we use the number of gradient descent steps to minimize $\mathcal{L}$ as a proxy, that is closely related to $-\log \delta$ when gradient descent has a linear convergence rate. We choose the number of steps to be the same in the two phases, for the sake of simplicity, even though it may not be optimal. We plot the evolution of the normalized error

$$\frac{\|\nabla_\omega - \widehat{\nabla}_\omega\|}{\|\nabla_\omega\|}$$

between the meta-gradient and the contrastive estimate (15) in Fig. S3. The qualitative behavior of this error, as a function of $\beta$ (Fig. S3A) and of number of steps (Fig. S3B), is accurately captured by Theorem 1 (compare with Fig. S1A and C).

We finish the study of this quadratic model by probing the $(\delta, \delta')$ space in a different way, by fixing the total number of steps and then modifying the allocation across the two phases (Fig. S3C). The best achievable error, as a function of $\beta$, decreases before some $\beta^*$ value and then increases, following the predictions of Theorem 1: too small $\beta$ values turn out to hurt performance when the solutions cannot be approximated arbitrarily well. Interestingly, the error plateaus for large $\beta$ values and the size of the plateau decreases with $\beta$ until reaching a critical value where it disappears. A conservative choice in practice is therefore to overestimate $\beta$, as it reduces the meta-gradient estimation sensitivity to a sub-optimal allocation, with only a minor degradation in the best achievable quality.

### S3.6   Verification the theoretical results on a simple hyperparameter optimization task

We now move to a more complicated setting that is closer to problems of practical interest, and in which we are not guaranteed that the assumptions of the theory hold. Still, it is simple enough such that we can calculate the exact value of the meta-gradient $\nabla_\theta$ using the analytical formula (24). This problem is a single-task regularization-strength learning problem [20, 21, 23, 52, 94] on the Boston housing dataset [98] (70% learning and 30% evaluation split). We study a nonlinear neural network model $f_\phi$ with a small hidden layer (20 neurons, hyperbolic tangent transfer function). The bilevel optimization problem we are solving here is the one we consider in Section 5.2, that is:

$$
\begin{aligned}
\min_\lambda \; & \frac{1}{|D^{\text{eval}}|} \sum_{(x,y) \in D^{\text{eval}}} l(f_{\phi_\lambda^*}(x), y) \\
\text{s.t. } \phi_\lambda^* \in & \arg\min_\phi \frac{1}{|D^{\text{learn}}|} \sum_{(x,y) \in D^{\text{learn}}} l(f_\phi(x), y) + \frac{1}{2} \sum_{i=1}^{|\phi|} \lambda_i \phi_i^2.
\end{aligned}
\tag{72}
$$

**Meta-gradient estimation error.**   We plot the normalized error between the meta-gradient estimate and its true value on Fig S4. The qualitative behavior closely matches the one we obtained for the quadratic analytical model in the last section, as well as the ones predicted by our theory.

**Comparison with other implicit gradient methods.**   We also use this problem to directly compare the meta-gradient approximation error made by our contrastive meta-learning rule (CML) to other implicit gradient methods, namely recurrent backpropagation (RBP) and conjugate gradient (CG). To make the comparison fair, we pick the hyperparameters that yield the smallest error for each method ($\beta$ and the parameters of the optimizer minimizing the second phase for CML, a scaling parameter for RBP, none for CG). Fig S5 characterizes the meta-gradient estimation errors by the different methods.

We first perfectly solve the first phase so that $\hat{\phi}_\lambda = \phi_\lambda^*$ and compare how efficient the second phase of those algorithms is. The Hessian of the learning loss $\partial_\phi^2 L^{\text{learn}}$ is positive definite as shown in Fig. S6A. The conjugate gradient method is therefore much more efficient than the other methods as its assumptions are met, and quickly reaches the numerical accuracy limit. Our rule compares favorably to recurrent backpropagation, even though the theoretical bound is weaker ($\sqrt{\delta'}$ for our rule compared to $\delta'$ for implicit methods [21]). A possible explanation comes from the fact that we are using Nesterov accelerated gradient descent for the second phase of our contrastive update, whereas the fixed point iteration of RBP is a form of gradient descent.

We repeat our analysis in the more realistic setting in which $\hat{\phi}_\lambda$ is not equal to $\phi_\lambda^*$. We use the same number of steps in the two phases (and the same estimate for $\hat{\phi}_\lambda$ and find that recurrent

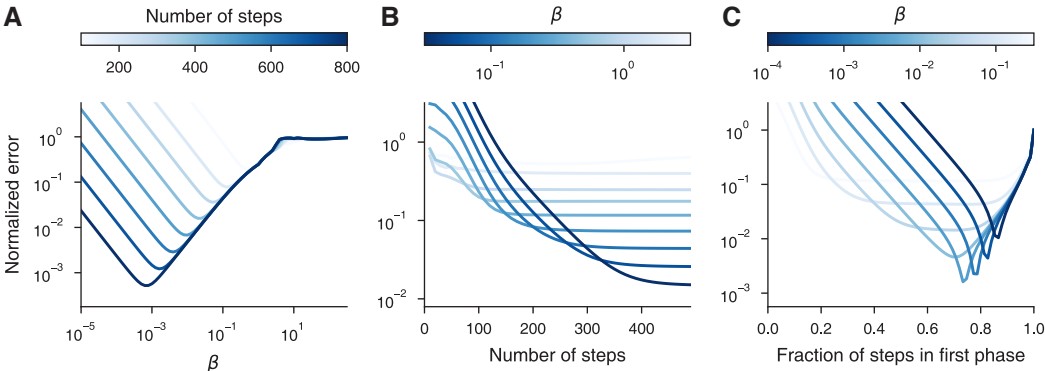

Figure S4: Empirical verification of the theoretical results on a regularization-strength learning problem on the Boston dataset. We plot the normalized error between the meta-gradient estimate $\widehat{\nabla}_\lambda$ and the true one $\nabla_\lambda$, as a function of $\beta$ (A), of the number of steps in the two phases (which is a proxy for $-\log\delta$ and $-\log\delta'$ used in the theory) (B), and as a function of the allocation of the computational resources between the two phases, the total number of steps being fixed to 750 (C).

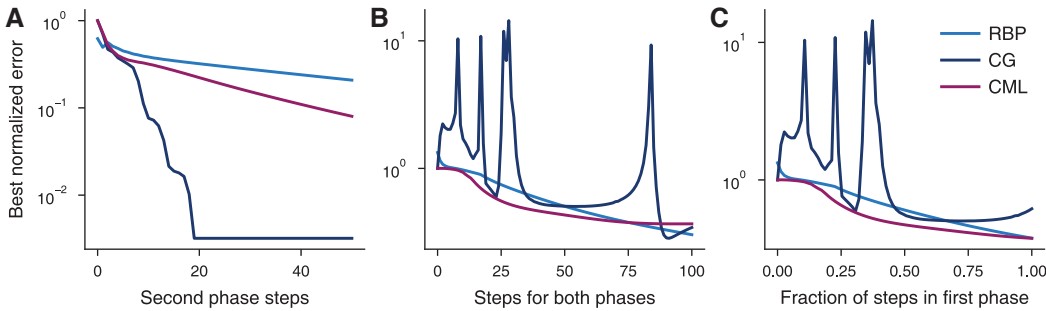

Figure S5: Comparison of the meta-gradient estimation errors provided our contrastive meta-learning rule (CML), recurrent backpropagation (RBP) and conjugate gradients (CG), on a regularization-strength learning problem on the Boston dataset. The hyperparameters for each value of the $x$ axis, such that the normalized error is minimized. (A) Error as a function of the number of steps in the second phase, the first phase being perfectly solved. (B) Error as a function of the number of steps performed in the two phase, which is fixed. (C) Error as a function of the fraction of steps in the first phase, the total number of steps for the two phases being fixed to 75.

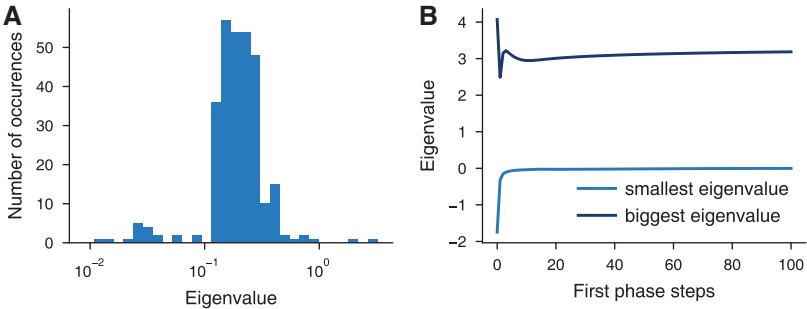

Figure S6: Eigenvalues of the Hessian of the learning loss $\partial_\phi^2 L^{\text{learn}}(\hat{\phi}_\lambda, \lambda)$ on the regularization-strength learning problem on the Boston dataset. (A) Spectrum of the Hessian when the first phase is perfectly solved, i.e., $\hat{\phi}_\lambda = \phi_\lambda^*$. (B) Smallest and biggest eigenvalue of the Hessian, as a function of the number of steps in the first phase. The higher the number of steps, the close $\hat{\phi}_\lambda$ is to $\phi_\lambda^*$.

backpropagation and our contrastive meta-learning rule improve their estimate of the meta-gradient as the number of steps for both phases increases, cf. Fig. S5.B. In contrast, conjugate gradient is unstable when the number of steps is low. In Fig. S6B, we check whether the needed assumptions are satisfied by plotting the smallest eigenvalue of the Hessian of the learning loss $\partial_\phi^2 L^{\text{learn}}(\hat{\phi}_\lambda, \lambda)$, as a function of the number of steps. We find that this eigenvalue is negative on the range of the number of steps we consider, the Hessian is therefore not positive definite so the conjugate gradient method cannot approximate $\mu$ well. We obtain the same qualitative behavior when we fix the total number of steps, and vary the faction of steps in the first phase, cf. Fig. S5C.

## S4  Experimental details

### S4.1  Supervised meta-optimization

**Task details.** For the supervised meta-optimization experiments we meta-learn parameter-wise l2-regularization strengths ($\omega = 0$) on the CIFAR-10 image classification task [48] starting each learning phase from a fixed neural network initialization. The dataset comprises 60000 32x32 RGB images divided into 10 classes, with 6000 images per class. We split the 50000 training images randomly in half to obtain a training set and a validation set for meta-learning and use the remaining 10000 test images for testing. In Tab. S2 we report additional results on the simpler MNIST image classification task [100] for which we use the same data splitting strategy.

Table S2: Meta-learning a per-synapse regularization strength meta-parameter (cf. Section 4.1) on MNIST. Average accuracies (acc.) $\pm$ s.e.m. over 10 seeds.

| Method | Validation acc. (%) | Test acc. (%) |
|---|---|---|
| T1-T2 | $98.70^{\pm00.08}$ | $97.63^{\pm00.03}$ |
| CG | $97.02^{\pm00.28}$ | $96.96^{\pm00.15}$ |
| RBP | $99.53^{\pm00.01}$ | $97.31^{\pm00.02}$ |
| CML | $99.45^{\pm00.16}$ | $97.92^{\pm00.11}$ |

**Additional results.** We perform an additional experiment investigating how the number of lower-level parameter updates affects the meta-learning performance of our method and comparison methods. We consider a simplified data regime for this experiment, using a random subset of 1000 examples of CIFAR-10 split into 50 samples for the learning loss and 950 samples for the evaluation loss which allows us to fit all samples into a single batch during learning and meta-learning. Results shown in Fig. S7 demonstrate that our contrastive meta-learning rule is able to fit the meta-parameters to the validation set across different number of lower-level parameter updates while competing methods require more updates to obtain similar performance. We found the conjugate gradient method (CG) to be unstable in this setting. To obtain these results, we tuned the hyperparameters for each method for each number of lower-level parameter updates.

**Architecture details.** For CIFAR-10 experiments we use a modified version of the classic LeNet-5 model [101] where we insert batch normalization layers [102] before each nonlinearity and replace the hyperbolic tangent nonlinearities with rectified linear units. For MNIST experiments we use a feedforward neural network with 5 hidden layers of size 256 and hyperbolic tangent nonlinearity.

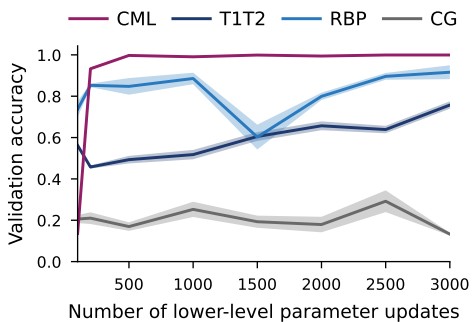

Figure S7: Dependence of the final validation accuracy on the number of lower-level parameter updates obtained after meta-optimizing per-synapse regularization strength meta-parameters on a subset of CIFAR-10. A random subset of 1000 examples from CIFAR-10 are split into 50 samples for the learning loss and 950 samples for the evaluation loss. Mean over 10 random seeds with error bars indicating $\pm 1$ s.e.m.

**Hyperparameters.** We perform a comprehensive random hyperparameter search for each method with the search space for CIFAR-10 experiments specified in Tab. S5 and the search space for MNIST experiments specified in Tab. S6.

In Fig. 1B, we furthermore investigate the interaction of $\beta$ and the number of first phase steps on the validation loss, keeping all other hyperparameters fixed. We compare it to the corresponding theoretical prediction visualized in Fig. S1B, for the case where $L^{\text{eval}}$ is independent of $\theta$ and $\delta'$ is fixed.

## S4.2 Few-shot image classification

**Task details.** We follow the standard experimental setup [18, 22, 103, 104] for our Omniglot [57] and miniImageNet [56] experiments.

**Additional results.** The results of related work reported in the main text (Tabs. 2 and 3) are taken from the original papers, except for Omniglot first-order MAML which is reported in ref. [58]. In Tab. S3 we provide results for additional 5-way Omniglot variants that are easier than the 20-way ones studied in the main text.

**Architecture details.** For Omniglot, we use max-pooling instead of stride in the convolutional layers, as we found the latter led to optimization instabilities, as previously reported [109]. We evaluate the statistics of batch normalization units [102] on the test set as in ref. [18], which yields a transductive classifier. More complex architectures whereby a second modulatory neural network which generates task-specific parameters is explicitly modeled [a hypernetwork; 110–112] can be easily accommodated into our framework, but here for simplicity we implement our top-down modulation model by taking advantage of existing batch normalization layers in our neural networks and consider the gain and shift parameters of these units as well as the synaptic weights and biases of the output layer as our task-specific parameters $\phi$.

**Optimization details.** We used the symmetric version of our contrastive rule for meta-learning and the Kaiming scheme for parameter initialization [113]. The task-specific learning and evaluation losses are both taken to be the cross-entropy with dataset splits into learning and evaluation data following the setup considered by Finn et al. [18]. In order to stabilize results, we used Polyak averaging [114] for the meta-parameters to compute final performance. Specifically, we started averaging meta-parameters after a certain number of meta-parameter updates (5 for Omniglot, 50 for miniImageNet). Note that the performance of the non-averaged meta-parameters performs only slightly differently averaged over iterations but is considerably more noisy.

**Hyperparameters.** We perform a comprehensive grid-search over hyperparameters with search ranges and optimal hyperparameters found reported in Tab. S7.

Table S3: Few-shot learning of Omniglot characters. We report results obtained with contrastive meta-learning for the synaptic and modulatory models. We present test set classification accuracy (%) averaged over 5 seeds $\pm$ std.

| Method | 5-way 1-shot | 5-way 5-shot | 20-way 1-shot | 20-way 5-shot |
|---|---|---|---|---|
| MAML [18] | $98.7^{\pm0.4}$ | $99.9^{\pm0.1}$ | $95.8^{\pm0.3}$ | $98.9^{\pm0.2}$ |
| First-order MAML [18] | $98.3^{\pm0.5}$ | $99.2^{\pm0.2}$ | $89.4^{\pm0.5}$ | $97.9^{\pm0.1}$ |
| Reptile [58] | $97.68^{\pm0.04}$ | $99.48^{\pm0.06}$ | $89.43^{\pm0.14}$ | $97.12^{\pm0.32}$ |
| iMAML [22] | $99.16^{\pm0.35}$ | $99.67^{\pm0.12}$ | $94.46^{\pm0.42}$ | $98.69^{\pm0.1}$ |
| CML (synaptic) | $98.11^{\pm0.34}$ | $99.49^{\pm0.16}$ | $94.16^{\pm0.12}$ | $98.06^{\pm0.26}$ |
| CML (modulatory) | $98.05^{\pm0.06}$ | $99.45^{\pm0.04}$ | $94.24^{\pm0.39}$ | $98.60^{\pm0.27}$ |

### S4.3 Few-shot regression in recurrent spiking network

**Task details.** We consider a standard sinusoidal 10-shot regression problem. For each task a sinusoid with random amplitude sampled uniformly from $[0.1, 5.0]$ and random phase sampled uniformly from $[0, \pi]$ is generated. 10 data points are drawn uniformly from the range $[-5, 5]$ both for the learning loss and for the evaluation loss.

**Architecture details.** We encode the input with a population of 100 neurons similar to Bellec et al. [61]. Each neuron $i$ has a Gaussian response field with the mean values $\mu_i$ evenly distributed in the range $[0, 1]$ across neurons and a fixed variance $\sigma^2 = 0.0002$. The firing probability of each neuron at a single time step is given by $p_i = \exp(\frac{-(\mu_i - z)^2}{2\sigma^2})$ where $z$ is the input value standardized to the range $[0, 1]$. We generate 20 time steps for each data point by sampling spikes from a Bernoulli distribution given the firing probabilities $p_i$ for each neuron.

We use a singe-layer recurrent spiking neural network with leaky integrate and fire neurons that follow the time-discretized dynamics with step size $\Delta t = 1.0$ (notation taken from Bellec et al. [59]):

$$h_j^{t+1} = \alpha h_j^t + \sum_{i \neq j} W_{ji}^{\text{rec}} z_i^t + \sum_i W_{ji}^{\text{in}} x_i^{t+1} - z_j^t v_{\text{th}} \tag{73}$$

$$z_j^t = \Theta(h_j^t - v_{\text{th}}) \tag{74}$$

$$y_k^{t+1} = \kappa y_k^t + \sum_j W_{kj}^{\text{out}} z_j^t \tag{75}$$

where $W^{\text{in}}, W^{\text{rec}}, W^{\text{out}}$ are the synaptic input, recurrent and output weights, $\alpha = \exp(-\frac{\Delta t}{\tau_{\text{hidden}}})$ and $\kappa = \exp(-\frac{\Delta t}{\tau_{\text{out}}})$ are decay factors with $\tau_{\text{hidden}} = \tau_{\text{out}} = 30.0$ , $v_{\text{th}} = 0.1$ is the threshold potential, and $\Theta(\cdot)$ denotes the Heaviside step function. The weights are initialized using the Kaiming normal scheme [113] and scaled down by a factor of $0.1, 0.01, 0.1$ for $W^{\text{in}}, W^{\text{rec}}, W^{\text{out}}$ respectively.

**Optimization details.** The weights are updated according to e-prop [59]:

$$\Delta W_{kj}^{\text{out}} \propto \sum_t (y_k^{*,t} - y_k^t) \sum_{t' \leq t} (\kappa^{t-t'} z_j^{t'}) \tag{76}$$

$$\Delta W_{ji}^{\text{rec}} \propto \sum_t (\sum_k W_{kj}^{\text{out}} (y_k^{*,t} - y_k^t)) \sum_{t' \leq t} (\kappa^{t-t'} h_j^{t'} \sum_{t'' \leq t'} (\alpha^{t'-t''} z_i^{t''})) \tag{77}$$

$$\Delta W_{ji}^{\text{in}} \propto \sum_t (\sum_k W_{kj}^{\text{out}} (y_k^{*,t} - y_k^t)) \sum_{t' \leq t} (\kappa^{t-t'} h_j^{t'} \sum_{t'' \leq t'} (\alpha^{t'-t''} x_i^{t''})) \tag{78}$$

The loss is computed as the mean-squared error between the target and the prediction given by the average output over time. Note that the output of the network is non-spiking. We add a regularization term to the loss that is computed as the mean squared difference between the average neuron firing rate and a target rate and decrease the learning rate for updating $W^{\text{out}}$ with e-prop by a factor of 0.1. We use the symmetric version of our contrastive rule to obtain meta-updates.

**Comparison methods.** We compare our method to a standard baseline where both fast parameter updates and slow meta-parameter updates are computed by backpropagating through the synaptic plasticity process (BPTT+BPTT) using surrogate gradients to handle spiking nonlinearities [60]. As this biologically-implausible process is computationally expensive, we restrict the number of update steps on the learning loss to 10 changes as done by prior work [18]. For a second comparison method, we compute the fast parameter updates using the e-prop update stated above and use backpropagation through 10 e-prop updates for meta-parameter updates (BPTT+e-prop).

**Hyperparameters.** For each method we employ an extensive random hyperparameter search over the search space defined in Tab. S8 using a meta-validation set to select the optimal set of hyperparameters.

## S4.4 Meta-reinforcement learning

**Task details.** The contextual wheel bandit was introduced by Riquelme et al. [64] to parametrize the task difficulty of a contextual bandit task in terms of its exploration-exploitation trade-off. Each task consists of a sequence of context coordinates $X$ randomly drawn from the unit circle and a scalar radius $\delta \in [0,1]$. The radius $\delta$ tiles the unit circle into a low-reward region and a high-reward region, see Fig. S8. If the current context lies within the low-reward region, $\|X\| \leq \delta$, all actions $a \in \{1,2,3,4\}$ return reward $r \sim \mathcal{N}(1.0, 0.01^2)$ except for the last action $a = 5$ which returns $r \sim \mathcal{N}(1.2, 0.01^2)$ and is thus optimal. If the current context lies within the high-reward region, one of the first four actions is optimal returning a high reward $r \sim \mathcal{N}(50.0, 0.01^2)$, the last arm still returns $r \sim \mathcal{N}(1.2, 0.01^2)$ and the remaining arms return $r \sim \mathcal{N}(1.0, 0.01^2)$. Which of the first four actions returns the high reward depends on the quadrant of the high-reward region in which the current context lies. Action 1 is optimal in the upper right quadrant, action 2 in the lower right quadrant, action 3 in the upper left quadrant and action 4 in the lower left quadrant.

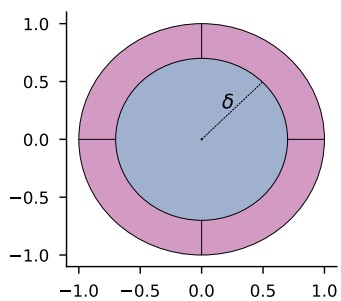

Figure S8: The wheel bandit task tiles the context space into an inner low-reward region (blue) and a high-reward outer rim (purple). Across tasks the radius $\delta$ of the inner low-reward region is varied. The high-reward region is divided into 4 quadrants, depending on which the optimal action changes.

**Additional results.** Following previous work [65, 66], we treat the contextual wheel bandit as a meta-learning problem. During meta-learning, we sample $M = 64$ different wheel tasks $\{\delta_i\}_{i=1}^{M}$, $\delta_i \sim \mathcal{U}(0,1)$ for each of which we sample a sequence of $N = 562$ random contexts, random actions and corresponding rewards $\{(X_j, a_j, r_j)\}_{j=1}^{N}$. We use 512 observations for the learning loss and 50 observations for the evaluation loss. Both the learning $l^{\text{learn}}$ and the evaluation loss $l^{\text{eval}}$ are measured as the mean-squared error between observed reward and the predicted value for the corresponding action. For each meta-learning step, we randomly sample from the $M$ tasks such that specific tasks may be encountered multiple times. After meta-learning, we evaluate the agent online on a long episode with 80000 contexts and track its cumulative reward relative to the cumulative reward obtained by an agent that chooses its actions at random. As done by Riquelme et al. [64], each action is initially explored twice before choosing actions according to the agent's policy. The extended results over more settings for $\delta$ can be seen in Tab. S4. Results for the NeuralLinear baseline reported here and in the main text are taken from the original paper [64].

**Architecture results.** For all methods including the synaptic consolidation and modulatory network model, we consider as a base a multilayer perceptron with ReLU nonlinearites and two hidden layers with 100 units. For the modulatory network model, each hidden unit is multiplied by a gain and shifted by a bias prior to applying the nonlinearity. The non-meta-learned baseline NeuralLinear we report from Riquelme et al. [64] additionally uses a Bayesian regression head for each action and applies Thompson sampling to choose actions.

**Optimization and evaluation details.** During online evaluation, we take the greedy action with respect to the predicted expected rewards on each context and store each observation $(X_j, a_j, r_j)$ in

Table S4: Cumulative regret on the wheel bandit problem for different values of $\delta$. Values are normalized by the cumulative regret of a uniformly random agent. Averages over 50 seeds $\pm$ s.e.m.

| $\delta$ | 0.5 | 0.7 | 0.9 | 0.95 | 0.99 |
|---|---|---|---|---|---|
| NeuralLinear [64] | $0.95^{\pm 0.02}$ | $1.60^{\pm 0.03}$ | $4.65^{\pm 0.18}$ | $9.56^{\pm 0.36}$ | $49.63^{\pm 2.41}$ |
| MAML | $0.45^{\pm 0.01}$ | $0.62^{\pm 0.03}$ | $1.02^{\pm 0.76}$ | $1.56^{\pm 0.62}$ | $15.21^{\pm 1.69}$ |
| CML (synaptic) | $0.40^{\pm 0.02}$ | $0.45^{\pm 0.01}$ | $0.82^{\pm 0.02}$ | $1.42^{\pm 0.07}$ | $12.27^{\pm 1.02}$ |
| CML (modulatory) | $0.42^{\pm 0.01}$ | $0.65^{\pm 0.03}$ | $1.83^{\pm 0.11}$ | $3.68^{\pm 0.59}$ | $16.46^{\pm 1.80}$ |

a replay buffer. This data is used to train the fast parameters every $t_f$ contexts for $t_s$ steps, where $t_f, t_s$ are hyperparameters tuned for every method.

**Hyperparameters.** We perform a comprehensive random hyperparameter search for each method with the search space specified in Tab. S9. Optimal parameters are selected on 5 validation tasks with $\delta = 0.95$.

Table S5: Hyperparameter search space for the supervised meta-optimization experiment on CIFAR-10. For all methods 500 samples were randomly drawn from the search space and Asynchronous HyperBand from ray tune [120] was used for scheduling with a grace period of 10. Best found parameters are marked in bold.

| Hyperparameter | CML | CG |
|---|---|---|
| batch_size | 500 | 500 |
| $\beta$ | $\{0.01, 0.03, 0.1, 0.3, 1.0, 3.0, \mathbf{10.0}\}$ | - |
| $\lambda$ | $\{10^{-5}, 10^{-4}, 10^{-3}, 10^{-2}, \mathbf{10^{-1}}\}$ | $\{\mathbf{10^{-5}}, 10^{-4}, 10^{-3}, 10^{-2}, 10^{-1}\}$ |
| lr_inner | $\{0.0001, \mathbf{0.0003}, 0.001, 0.003, 0.01, 0.03, 0.1\}$ | $\{0.0001, 0.0003, \mathbf{0.001}, 0.003, 0.01, 0.03, 0.1\}$ |
| lr_nudged | $\{\mathbf{0.0001}, 0.0003, 0.001, 0.003, 0.01, 0.03, 0.1\}$ | - |
| lr_outer | $\{0.0001, 0.0003, 0.001, 0.003, 0.01, \mathbf{0.03}, 0.1\}$ | $\{0.0001, 0.0003, \mathbf{0.001}, 0.003, 0.01, 0.03, 0.1\}$ |
| optimizer_inner | $\{$adam, $\mathbf{sgd\_nesterov\_0.9}\}$ | $\{$adam, $\mathbf{sgd\_nesterov\_0.9}\}$ |
| optimizer_outer | adam | adam |
| steps_cg | - | $\{\mathbf{100}, 500, 1000, 2000\}$ |
| steps_inner | $\{2000, 3000, \mathbf{5000}\}$ | $\{\mathbf{2000}, 3000, 5000\}$ |
| steps_nudged | $\{100, \mathbf{200}, 500\}$ | - |
| steps_outer | 100 | 100 |

| Hyperparameter | NSA | T1T2 |
|---|---|---|
| batch_size | 500 | 500 |
| $\lambda$ | $\{\mathbf{10^{-5}}, 10^{-4}, 10^{-3}, 10^{-2}, 10^{-1}\}$ | $\{10^{-5}, \mathbf{10^{-4}}, 10^{-3}, 10^{-2}, 10^{-1}\}$ |
| lr_inner | $\{0.0001, 0.0003, \mathbf{0.001}, 0.003, 0.01, 0.03, 0.1\}$ | $\{0.0001, 0.0003, \mathbf{0.001}, 0.003, 0.01, 0.03, 0.1\}$ |
| lr_outer | $\{0.0001, 0.0003, 0.001, \mathbf{0.003}, 0.01, 0.03, 0.1\}$ | $\{0.0001, 0.0003, \mathbf{0.001}, 0.003, 0.01, 0.03, 0.1\}$ |
| nsa_alpha | $\{0.000001, 0.000003, 0.00001, 0.00003, 0.0001, \mathbf{0.0003}\}$ | - |
| optimizer_inner | $\{$adam, $\mathbf{sgd\_nesterov\_0.9}\}$ | $\{$adam, $\mathbf{sgd\_nesterov\_0.9}\}$ |
| optimizer_outer | adam | adam |
| steps_inner | $\{\mathbf{2000}, 3000, 5000\}$ | $\{2000, \mathbf{3000}, 5000\}$ |
| steps_nsa | 500 | - |
| steps_outer | 100 | 100 |

| Hyperparameter | TBPTL | no-meta |
|---|---|---|
| batch_size | 500 | 500 |
| $\lambda$ | $\{10^{-5}, \mathbf{10^{-4}}, 10^{-3}, 10^{-2}, 10^{-1}\}$ | $\{10^{-5}, 10^{-4}, 10^{-3}, \mathbf{10^{-2}}, 10^{-1}\}$ |
| lr_inner | $\{0.0001, 0.0003, \mathbf{0.001}, 0.003, 0.01, 0.03, 0.1\}$ | $\{0.0001, 0.0003, \mathbf{0.001}, 0.003, 0.01, 0.03, 0.1\}$ |
| lr_outer | $\{0.0001, 0.0003, \mathbf{0.001}, 0.003, 0.01, 0.03, 0.1\}$ | - |
| optimizer_inner | $\{$adam, $\mathbf{sgd\_nesterov\_0.9}\}$ | $\{$adam, $\mathbf{sgd\_nesterov\_0.9}\}$ |
| optimizer_outer | adam | - |
| steps_inner | $\{2000, \mathbf{3000}, 5000\}$ | 5000 |
| steps_outer | 100 | 0 |

Table S6: Hyperparameter search space for the supervised meta-optimization experiment on MNIST. For all methods 500 samples were randomly drawn from the search space and Asynchronous HyperBand from ray tune [120] was used for scheduling with a grace period of 10. Best found parameters are marked in bold.

| Hyperparameter | CML | CG |
|---|---|---|
| batch_size | 500 | 500 |
| $\beta$ | {0.01, 0.03, 0.1, 0.3, 1.0, **3.0**, 10.0} | - |
| $\lambda$ | {**0.00001**, 0.0001, 0.001, 0.01, 0.1} | {0.00001, **0.0001**, 0.001, 0.01, 0.1} |
| lr_inner | {0.0001, **0.0003**, 0.001, 0.003, 0.01, 0.03, 0.1} | {**0.0001**, 0.0003, 0.001, 0.003, 0.01, 0.03, 0.1} |
| lr_nudged | {0.0001, **0.0003**, 0.001, 0.003, 0.01, 0.03, 0.1} | - |
| lr_outer | {**0.0001**, 0.0003, 0.001, 0.003, 0.01, 0.03, 0.1} | {0.0001, **0.0003**, 0.001, 0.003, 0.01, 0.03, 0.1} |
| optimizer_inner | {**adam**, sgd_nesterov_0.9} | {**adam**, sgd_nesterov_0.9} |
| optimizer_outer | adam | adam |
| steps_cg | - | {**100**, 250, 500, 1000, 2000} |
| steps_inner | 2000 | 2000 |
| steps_nudged | {100, **200**, 500} | - |
| steps_outer | 100 | 100 |

| Hyperparameter | NSA | T1T2 |
|---|---|---|
| batch_size | 500 | 500 |
| $\lambda$ | {0.00001, **0.0001**, 0.001, 0.01, 0.1} | {**0.00001**, 0.0001, 0.001, 0.01, 0.1} |
| lr_inner | {0.0001, 0.0003, **0.001**, 0.003, 0.01, 0.03, 0.1} | {0.0001, 0.0003, 0.001, 0.003, **0.01**, 0.03, 0.1} |
| lr_outer | {0.0001, 0.0003, 0.001, 0.003, **0.01**, 0.03, 0.1} | {**0.0001**, 0.0003, 0.001, 0.003, 0.01, 0.03, 0.1} |
| nsa_alpha | {0.000001, 0.000003, 0.00001, 0.00003, 0.0001, 0.0003, **0.001**, 0.003} | - |
| optimizer_inner | {adam, **sgd_nesterov_0.9**} | {adam, **sgd_nesterov_0.9**} |
| optimizer_outer | adam | adam |
| steps_inner | 2000 | 2000 |
| steps_nsa | 200 | - |
| steps_outer | 100 | 100 |

Table S7: Hyperparameter search space for the few-shot image classification experiments on Omniglot and miniImageNet. Best found parameters are marked in bold.

| Hyperparameter | Omni-5W-1s | Omni-5W-5s | Omni-20W-1s | Omni-20W-5s | miniImageNet |
|---|---|---|---|---|---|
| batch_size | 32 | 32 | 16 | 16 | 4 |
| $\beta$ | {0.01, 0.03,**0.1**, 0.3, 1.} | {0.01, 0.03,**0.1**, 0.3, 1.} | {0.01, **0.03**, 0.1, 0.3, 1.} | {0.01, 0.03, **0.1**, 0.3, 1.} | {**0.01**, 0.03, 0.1, 0.3, 1.} |
| $\lambda$ | {0.1, 0.25,**0.5**} | {**0.1**, 0.25, 0.5} | {0.1, 0.25, **0.5**} | {0.1, **0.25**, 0.5} | {0.1, 0.25, **0.5**} |
| lr_inner | 0.01 | 0.01 | 0.01 | 0.01 | 0.01 |
| lr_outer | {**0.01**, 0.001 } | {**0.01**, 0.001 } | {**0.01**, 0.001 } | {**0.01**, 0.001 } | 0.001 |
| optimizer_inner | gd_nesterov_0.9 | gd_nesterov_0.9 | gd_nesterov_0.9 | gd_nesterov_0.9 | gd_nesterov_0.9 |
| optimizer_outer | adam | adam | adam | adam | adam |
| steps_inner | {50, 100, 150, **200**} | {50, 100, 150, **200**} | {50, 100, 150, **200**} | {50, 100, 150, **200**} | {50, **100**, 150, 200} |
| steps_nudged | {50, **100**, 150, 200} | {50, **100**, 150, 200} | {50, **100**, 150, 200} | {50, 100, 150, **200**} | {25, 50, **75**, 100} |
| steps_outer | 3750 | 3750 | 3750 | 3750 | 25000 |

Table S8: Hyperparameter search space for the sinusoidal fewshot regression experiment. For all methods 500 samples were randomly drawn from the search space and the Asynchronous HyperBand scheduler from ray tune was used with a grace period of 10 [120]. Best found parameters are marked in bold.

| Hyperparameter | CML + e-prop |
|---|---|
| activity_reg_strength | $\{10^{-1}, 10^{-2}, 10^{-3}, 10^{-4}, \mathbf{10^{-5}}, 10^{-6}\}$ |
| activity_reg_target | $\{0.05, 0.1, \mathbf{0.2}\}$ |
| batch_size | $\{1, 5, \mathbf{10}\}$ |
| $\beta$ | $\{0.01, 0.03, 0.1, 0.3, 1.0, \mathbf{3.0}, 10.0\}$ |
| $\lambda$ | $\{10^0, 10^{-1}, \mathbf{10^{-2}}, 10^{-3}, 10^{-4}, 10^{-5}, 10^{-6}\}$ |
| lr_inner | $\{0.0001, 0.0003, \mathbf{0.001}, 0.003, 0.01, 0.03, 0.1\}$ |
| lr_nudged | $\{0.0001, 0.0003, 0.001, \mathbf{0.003}, 0.01, 0.03, 0.1\}$ |
| lr_outer | $\{0.0001, 0.0003, 0.001, \mathbf{0.003}, 0.01, 0.03, 0.1\}$ |
| meta_batch_size | $\{1, 10, \mathbf{25}\}$ |
| optimizer_inner | $\{\mathbf{adam}, \text{sgd\_nesterov\_0.9}\}$ |
| optimizer_outer | adam |
| steps_inner | 500 |
| steps_nudged | $\{50, \mathbf{100}, 200\}$ |
| steps_outer | 1000 |

| Hyperparameter | BPTT + e-prop | BPTT + BPTT | TBPTL + e-prop |
|---|---|---|---|
| activity_reg_strength | $\{10^{-1}, \mathbf{10^{-2}}, 10^{-3}, 10^{-4}, 10^{-5}, 10^{-6}\}$ | $\{10^{-1}, 10^{-2}, 10^{-3}, 10^{-4}, 10^{-5}, \mathbf{10^{-6}}\}$ | $\{10^{-1}, 10^{-2}, 10^{-3}, 10^{-4}, 10^{-5}, \mathbf{10^{-6}}\}$ |
| activity_reg_target | $\{\mathbf{0.05}, 0.1, 0.2\}$ | $\{0.05, \mathbf{0.1}, 0.2\}$ | $\{0.05, \mathbf{0.1}, 0.2\}$ |
| batch_size | $\{\mathbf{1}, 5, 10\}$ | $\{1, 5, \mathbf{10}\}$ | $\{\mathbf{1}, 5, 10\}$ |
| lr_inner | $\{\mathbf{0.0001}, 0.0003, 0.001, 0.003, 0.01, 0.03, 0.1\}$ | $\{0.0001, \mathbf{0.0003}, 0.001, 0.003, 0.01, 0.03, 0.1\}$ | $\{\mathbf{0.0001}, 0.0003, 0.001, 0.003, 0.01, 0.03, 0.1\}$ |
| lr_outer | $\{0.0001, \mathbf{0.0003}, 0.001, 0.003, 0.01, 0.03, 0.1\}$ | $\{0.0001, \mathbf{0.0003}, 0.001, 0.003, 0.01, 0.03, 0.1\}$ | $\{0.0001, 0.0003, \mathbf{0.001}, 0.003, 0.01, 0.03, 0.1\}$ |
| meta_batch_size | $\{1, \mathbf{10}, 25\}$ | $\{\mathbf{1}, 10, 25\}$ | $\{\mathbf{1}, 10, 25\}$ |
| optimizer_inner | $\{\text{adam}, \mathbf{sgd\_nesterov\_0.9}\}$ | $\{\text{adam}, \mathbf{sgd\_nesterov\_0.9}\}$ | sgd |
| optimizer_outer | adam | adam | adam |
| steps_inner | 10 | 10 | 500 |
| steps_outer | 1000 | 1000 | 1000 |

Table S9: Hyperparameter search space for the wheel bandit experiment. For all methods 1000 samples were randomly drawn from the search space. Best found parameters are marked in bold.

| Hyperparameter | CML (synaptic) | CML (modulatory) | MAML |
|---|---|---|---|
| batch_size | 512 | 512 | 512 |
| $\beta$ | {0.01, 0.03, 0.1, **0.3**, 1.0, 3.0, 10.0} | {0.01, 0.03, 0.1, 0.3, 1.0, 3.0, **10.0**} | - |
| $\lambda$ | {$10^{-6}, 10^{-5}, \ldots, \mathbf{10^3}$} | - | - |
| lr_inner | {**0.0001**, 0.0003, 0.001, 0.003, 0.01, 0.03, 0.1} | {0.0001, **0.0003**, 0.001, 0.003, 0.01, 0.03, 0.1} | {0.0001, 0.0003, 0.001, 0.003, **0.01**, 0.03, 0.1} |
| lr_nudged | {0.0001, 0.0003, 0.001, 0.003, 0.01, **0.03**, 0.1} | {**0.0001**, 0.0003, 0.001, 0.003, 0.01, 0.03, 0.1} | - |
| lr_online | {**0.0001**, 0.0003, 0.001, 0.003, 0.01, 0.03, 0.1} | {**0.0001**, 0.0003, **0.001**, 0.003, 0.01, 0.03, 0.1} | {0.0001, 0.0003, **0.001**, 0.003, 0.01, 0.03, 0.1} |
| lr_outer | {0.0001, 0.0003, 0.001, 0.003, 0.01, **0.03**, 0.1} | {0.0001, 0.0003, 0.001, 0.003, 0.01, **0.03**, 0.1} | {0.0001, 0.0003, 0.001, 0.003, 0.01, 0.03, **0.1**} |
| meta_batch_size | {**8**, 16, 32} | {8, **16**, 32} | {8, 16, **32**} |
| optimizer_inner | adam | {adam, **sgd**, sgd_nesterov_0.9} | sgd |
| optimizer_online | adam | {adam, sgd, **sgd_nesterov_0.9**} | sgd |
| optimizer_outer | adam | {**adam**, adamw} | adam |
| steps_inner | {100, **250**, 500, 1000} | {100, 250, 500, **1000**} | {5, **10**, 50, 100} |
| steps_nudged | {**100**, 250, 500, 1000} | {**100**, 250, 500, 1000} | - |
| steps_outer | 6400 | 6400 | 6400 |
| $t_f$ | {20, **50**, 100} | {20, 50, **100**} | {20, 50, **100**} |
| $t_s$ | {50, 100, **250**, 500, 1000} | {50, 100, 250, **500**, 1000} | {5, 10, 50, 100} |

# S5 Additional details

## S5.1 Compute resources

We used Linux workstations with 2 Nvidia RTX 3090 and 4 Nvidia RTX 3070 GPUs during development and conducted hyperparameter searches and larger experiments on up to 3 Linux servers with 8 Nvidia RTX 3090 GPUs with 24 GB memory each. Most of the experiments and corresponding hyperparameter scans presented take less than a few hours to complete on a single server. The more challenging recurrent spiking network and miniImageNet experiments require approximately 2-5 days to complete. During development we conducted many more hyperparameter scans over the course of several months.

## S5.2 Software and libraries

For the results produced in this paper we relied on free and open-source software. We implemented our experiments in Python using PyTorch [121, BSD-style license], JAX [122, Apache License 2.0], Ray [120, Apache License 2.0] and NumPy [123, BSD-style license]. For the visual few-shot classification dataset splits we used the Torchmeta library [124, MIT license] and for the generation of plots we used matplotlib [125, BSD-style license].

## S5.3 Datasets

We conducted our experiments with the public domain datasets Boston housing [98, MIT License], MNIST [100, GNU GPL v3.0], Omniglot [57] (MIT license), miniImageNet [56] (custom MIT/ImageNet license) and CIFAR-10 (MIT license) [48].

## Supplementary References

[29] Benjamin Scellier and Yoshua Bengio. Equilibrium propagation: bridging the gap between energy-based models and backpropagation. *Frontiers in Computational Neuroscience*, 11, 2017.

[30] Benjamin Scellier. *A deep learning theory for neural networks grounded in physics*. PhD Thesis, Université de Montréal, 2021.

[24] Asen L. Dontchev and R. Tyrrell Rockafellar. *Implicit Functions and Solution Mappings*. Springer, NY, 2009.

[87] Axel Laborieux, Maxence Ernoult, Benjamin Scellier, Yoshua Bengio, Julie Grollier, and Damien Querlioz. Scaling equilibrium propagation to deep convnets by drastically reducing its gradient estimator bias. *Frontiers in Neuroscience*, 14, 2021.

[53] Jelena Luketina, Mathias Berglund, Klaus Greff, and Tapani Raiko. Scalable gradient-based tuning of continuous regularization hyperparameters. In *International Conference on Machine Learning*, 2016.

[51] Renjie Liao, Yuwen Xiong, Ethan Fetaya, Lisa Zhang, KiJung Yoon, Xaq Pitkow, Raquel Urtasun, and Richard Zemel. Reviving and improving recurrent back-propagation. In *International Conference on Machine Learning*, 2018.

[23] Jonathan Lorraine, Paul Vicol, and David Duvenaud. Optimizing millions of hyperparameters by implicit differentiation. In *International Conference on Artificial Intelligence and Statistics*, 2020.

[91] C. Goutte and J. Larsen. Adaptive regularization of neural networks using conjugate gradient. In *Proceedings of the IEEE International Conference on Acoustics, Speech and Signal Processing*, 1998.

[22] Aravind Rajeswaran, Chelsea Finn, Sham Kakade, and Sergey Levine. Meta-learning with implicit gradients. In *Advances in Neural Information Processing Systems*, 2019.

[54] Amirreza Shaban, Ching-An Cheng, Nathan Hatch, and Byron Boots. Truncated back-propagation for bilevel optimization. In *International Conference on Artificial Intelligence and Statistics*, 2019.

[94] David J. C. MacKay. A practical Bayesian framework for backpropagation networks. *Neural Computation*, 4(3), 1992.

[20] Yoshua Bengio. Gradient-based optimization of hyperparameters. *Neural Computation*, 12(8): 1889–1900, 2000.

[52] Chuan-sheng Foo, Chuong B. Do, and Andrew Y. Ng. Efficient multiple hyperparameter learning for log-linear models. In *Advances in Neural Information Processing Systems*, 2007.

[21] Fabian Pedregosa. Hyperparameter optimization with approximate gradient. In *International Conference on Machine Learning*, 2016.

[98] David Harrison Jr. and Daniel L. Rubinfeld. Hedonic housing prices and the demand for clean air. *Journal of Environmental Economics and Management*, 5(1):81–102, 1978.

[48] Alex Krizhevsky. Learning multiple layers of features from tiny images. Technical report, 2009.

[100] Yann LeCun. The MNIST database of handwritten digits. *Available at http://yann. lecun. com/exdb/mnist*, 1998.

[101] Yann LeCun, Léon Bottou, Yoshua Bengio, and Patrick Haffner. Gradient-based learning applied to document recognition. *Proceedings of the IEEE*, 86(11):2278–2324, 1998.

[102] Sergey Ioffe and Christian Szegedy. Batch normalization: accelerating deep network training by reducing internal covariate shift. In *International Conference on Machine Learning*, 2015.

[103] Adam Santoro, Sergey Bartunov, Matthew Botvinick, Daan Wierstra, and Timothy Lillicrap. Meta-learning with memory-augmented neural networks. In *International Conference on Machine Learning*, 2016.

[104] Oriol Vinyals, Charles Blundell, Timothy P. Lillicrap, Koray Kavukcuoglu, and Daan Wierstra. Matching networks for one shot learning. In *Advances in Neural Information Processing Systems*, 2016.

[18] Chelsea Finn, Pieter Abbeel, and Sergey Levine. Model-agnostic meta-learning for fast adaptation of deep networks. In *International Conference on Machine Learning*, 2017.

[57] Brenden M. Lake, Ruslan Salakhutdinov, Jason Gross, and Joshua B. Tenenbaum. One shot learning of simple visual concepts. In *Proceedings of the Annual Meeting of the Cognitive Science Society*, 2011.

[56] Sachin Ravi and Hugo Larochelle. Optimization as a model for few-shot learning. In *International Conference on Learning Representations*, 2016.

[58] Alex Nichol, Joshua Achiam, and John Schulman. On first-order meta-learning algorithms. *arXiv preprint arXiv:1803.02999*, 2018.

[109] Antreas Antoniou, Harrison Edwards, and Amos Storkey. How to train your MAML. In *International Conference on Learning Representations*, 2019.

[110] David Ha, Andrew Dai, and Quoc V. Le. HyperNetworks. In *International Conference on Learning Representations*, 2017.

[111] Andrei A. Rusu, Dushyant Rao, Jakub Sygnowski, Oriol Vinyals, Razvan Pascanu, Simon Osindero, and Raia Hadsell. Meta-learning with latent embedding optimization. In *International Conference on Learning Representations*, 2019.

[112] Dominic Zhao, Seijin Kobayashi, João Sacramento, and Johannes von Oswald. Meta-learning via hypernetworks. In *Workshop on Meta-Learning at NeurIPS*, 2020.

[113] Kaiming He, Xiangyu Zhang, Shaoqing Ren, and Jian Sun. Delving deep into rectifiers: surpassing human-level performance on ImageNet classification. In *Proceedings of the IEEE International Conference on Computer Vision*, 2015.

[114] Boris T. Polyak and Anatoli B. Juditsky. Acceleration of stochastic approximation by averaging. *SIAM Journal on Control and Optimization*, 30(4):838–855, 1992.

[59] Guillaume Bellec, Franz Scherr, Anand Subramoney, Elias Hajek, Darjan Salaj, Robert Legenstein, and Wolfgang Maass. A solution to the learning dilemma for recurrent networks of spiking neurons. *Nature Communications*, 11(1):3625, 2020.

[60] Emre O. Neftci, Hesham Mostafa, and Friedemann Zenke. Surrogate gradient learning in spiking neural networks: bringing the power of gradient-based optimization to spiking neural networks. *IEEE Signal Processing Magazine*, 36(6):51–63, 2019.

[65] Marta Garnelo, Jonathan Schwarz, Dan Rosenbaum, Fabio Viola, Danilo J. Rezende, S. M. Ali Eslami, and Yee Whye Teh. Neural processes. *arXiv preprint arXiv:1807.01622*, 2018.

[66] Sachin Ravi and Alex Beatson. Amortized Bayesian meta-learning. In *International Conference on Learning Representations*, 2019.

[64] Carlos Riquelme, George Tucker, and Jasper Snoek. Deep Bayesian bandits showdown: an empirical comparison of Bayesian deep networks for Thompson sampling. In *International Conference on Learning Representations*, 2018.

[120] Richard Liaw, Eric Liang, Robert Nishihara, Philipp Moritz, Joseph E. Gonzalez, and Ion Stoica. Tune: A research platform for distributed model selection and training. *arXiv preprint arXiv:1807.05118*, 2018.

[121] Adam Paszke, Sam Gross, Francisco Massa, Adam Lerer, James Bradbury, Gregory Chanan, Trevor Killeen, Zeming Lin, Natalia Gimelshein, Luca Antiga, Alban Desmaison, Andreas Kopf, Edward Yang, Zachary DeVito, Martin Raison, Alykhan Tejani, Sasank Chilamkurthy, Benoit Steiner, Lu Fang, Junjie Bai, and Soumith Chintala. PyTorch: an imperative style, high-performance deep learning library. In *Advances in Neural Information Processing Systems*, 2019.

[122] James Bradbury, Roy Frostig, Peter Hawkins, Matthew James Johnson, Chris Leary, Dougal Maclaurin, George Necula, Adam Paszke, Jake VanderPlas, Skye Wanderman-Milne, and Qiao Zhang. JAX: composable transformations of Python+NumPy programs, 2018. URL `http://github.com/google/jax`.

[123] Charles R. Harris, K. Jarrod Millman, Stéfan J. van der Walt, Ralf Gommers, Pauli Virtanen, David Cournapeau, Eric Wieser, Julian Taylor, Sebastian Berg, Nathaniel J. Smith, Robert Kern, Matti Picus, Stephan Hoyer, Marten H. van Kerkwijk, Matthew Brett, Allan Haldane, Jaime Fernández del Río, Mark Wiebe, Pearu Peterson, Pierre Gérard-Marchant, Kevin Sheppard, Tyler Reddy, Warren Weckesser, Hameer Abbasi, Christoph Gohlke, and Travis E. Oliphant. Array programming with NumPy. *Nature*, 585(7825):357–362, 2020.

[124] Tristan Deleu, Tobias Würfl, Mandana Samiei, Joseph Paul Cohen, and Yoshua Bengio. Torchmeta: A meta-learning library for PyTorch. *arXiv preprint arXiv:1909.06576*, 2019.

[125] J. D. Hunter. Matplotlib: A 2D graphics environment. *Computing in Science & Engineering*, 9(3):90–95, 2007.