# OpenReview forum: "A contrastive rule for meta-learning"
_NeurIPS.cc/2022/Conference — NeurIPS 2022 Accept_

### Official Review · Reviewer_YWLg · 2022-07-08

**Rating:** 6
**Confidence:** 3
**Soundness:** 3 good
**Presentation:** 3 good
**Contribution:** 3 good

**Summary:**

Paper addresses the problem of performing meta-learning when facing different tasks by introducing a general-purpose rule of contrastive kind that aims on biological plausibility. The main idea of contrastive meta-learning here is to run the underlying algorithm twice: once by learning the original presented task  and once to learn what the authors call "augmented" version of the task that includes the meta-objective. From this perspective, authors see their approach as generalization of classical Hebbian contrastive learning to meta-learning frame. When arguing for biological plausibility, authors state that
1. the rule runs forward in time, avoiding backtracking through synaptic modifications in reverse time order,
2. the rule does not evaluate second derivatives, avoiding using information non-local to an updated parameter
3. implementing the rule requires only temporarily storing one intermediate state

Authors also derive an expression for estimating meta-gradient approximation error that arises by applying the rule, and state that meta-gradients can be consequently approximated as accurately as need, by investing more resources spent in learning and by controlling the nudging strength $\beta$ that controls contribution of the meta-objective term in the total loss.

To show concrete implementations of their rule, the authors show two distinct versions using different meta-parameters. One is based on the complex synapse idea, where the synapse is allowed to have further internal variables apart from the effective synaptic weight. The other uses notion of fast top-down modulation, which affects two additional meta-parameters per neuron - its adaptive multiplicative gain $g$ and adaptive threshold $b$.

The authors then benchmark their proposed contrastive meta-learning algorithm using the two proposed implementations against alternatives using a number of tasks: supervised meta-learning on CIFAR-10; supervised few-shot meta-learning on miniImageNet and Omniglot; supervised meta-learning in a recurrent spiking network using sinusiodal functions dataset; and meta-learning in reward-based setting using wheel bandit task.

On the basis of the benchmarks, authors state that their contrastive meta-learning rule shows comparable of better performance than the alternative counterparts. Authors conclude that their biologically plausible rule has thus competitive or better performance than meta-learning methods that have to rely on non-local information. Finally, in the discussion section, authors elaborate on how the two phases of learning and the augmented version of the tasks that are required by their rule may be instantiated in the brain networks, taking hippocampus as possible candidate structure responsible for coordinating this learning form.

**Questions:**

1. It seems that the rule relies on knoweldge of the task identity $\tau$. It would be good to discuss how learning algorithm may obtain this knoweldge in biological plausible way and still remain local.
2. Does the training assume i.i.d shuffled data? If yes, how is that executed in biological plausible way? If no, what makes the algorithm perform continual learning?
3. Why not combining both complex synapse and top-down modulation implementation in one rule? It seems both mechanisms are co-existing in the brain.
4. What about benchmarking the algorithm on more realistic multi-task datasets? For instance, the standard ATARI game suit or MetaDataset (https://github.com/google-research/meta-dataset) would give a proper multi-task setting and allow for more generalizable comparision to other approaches.
5. In the discussion, you point to hippocampus as a "control" structure that organizes contrastive learning - creating augmented task version and coordinating original and augmented task learning. It would be good to give a more grounded explanation - why necessary hippocampus (and not another structure, say basal ganglia or pre-frontal cortex, or insula, that also have access to most of cortical networks) and how it may subserve these functions? How sleep related activity may be relevant here, given that there will be a large time lag between learning episodes and interference with all possible tasks that happened during this period of time?
6. Paper mentions one intermediate state that has to be temporary buffered after original task learning was executed, until augmented task learning has been performed. Is this state local to the synapses? Where is it in the update equations (is it $\phi_{\theta,0,\tau}$)? It would be good to make this explicit for the reader. What is the biological plausible way to perform this short-term storage?
7. It is stated that meta-gradient approximation error can be made arbitrary small by either running further learning or decreasing nudging factor $\beta$. What may be a biologically plausible way to tune the nudging factor?
8. It is mentioned that in case of implementing the rule by top-down modulation, the partial derivatives of the augmented loss function will be computed by automatic differentiation using standard backprop. To rescue biological plausibility, further mechanisms are invoked to replace backprop : prediction error neural subpopulations, dendritic error representations, or usage of equilibrium propagation. Would the same mechanisms then act for replacing backpropagation in general also for the original task learning, or is the idea here that those will be acting only for learning on augmented task?
9. In general, how would usual approaches to replace backpropagation with local rules (again, for instance equilibrium propagation) relate to the proposed contrastive rule - are they complementary, or belong to the same class of methods, only applied for meta-learning in this case?


**Limitations:**

One limitation of the study that would be good to mention explicitly is the very small scale of conducted experiments (both network size and data size wise), compared to current state-of-the-art in deep learning.

Authors emphasize strongly benefits of top-down modulation based rule implementation. I think it would be good to cite corresponding classical works (eg Schmidhuber in 90s, Fast Weight Programmers, see for example discussion in Going Beyond Linear Transformers with Recurrent Fast Weight Programmers, https://proceedings.neurips.cc/paper/2021/hash/3f9e3767ef3b10a0de4c256d7ef9805d-Abstract.html ) that deal with fast slow learners, describing the same way of fast weight modulation by a slowly learning network (dual architecture as mentioned by authors), as alternative to slow weight change for task learning.

**Strengths And Weaknesses:**

## Strengths

The paper introduces a novel take on biologically plausible meta-learning formulating a rule that attempts to use only locally available information for parameter updates. It uses the idea of contrastive learning in novel context by proposing to contrast between two different learning episodes based on the same task. I think this is an interesting idea to elaborate on in general - contrasting on the level of task instead of usual notion of contrasting on the level of data samples within a task. Authors motivate their rule in clear and conscise manner, emphasizing the point of biological plausibility, resulting locality and the differences to conventional meta-learning approaches. While there is some amount of work being done on deriving local plasticity rules that may replace backprop in standard learning, to my knowledge it is still rare to treat the same problem in meta-learning frame, and therefore significant for the community. Theoretical derivation of meta-gradient approximation error and the notion that approximation can be in principle made arbitrary accurate adds further value.

## Weaknesses
While the contrastive meta-learning rule aims on handling a setting with multiple various tasks streaming in, it is not clear for me after reading the manuscript how such a multi task setting should be handled in biologically plausible manner, which is one of central claims of the paper. It seems that rule cannot work when task do not have clear boundaries and needs some mechanism that identifies a task while it is learned. Further it is not clear what happens if there are concurrent tasks present at the same time.

Another point is handling of catastrophical forgetting and non-shuffled data: the authors do not elaborate whether their rule works under i.i.d assumption with respect to the tasks or whether it can also handle non-shuffled data (which would be another pre-requisite for biological plausible setting).

In the benchmarking, there is in my opinion lack of scenarios that test meta-learning in proper multi-task frame, where tasks have indeed different structure. So, meta-learning on CIFAR-10 is actually dealing with one task, while miniImageNet and Omniglot are used in rather classical few-shot scenarios. where the task is again the same across episodes. Other benchmarking tasks - sinusoidal function regression and wheel bandit - can be arguably considered as toy tasks, and it is not clear whether benchmarking on those can give us generic statements about the power of the algorithm.

My impression is also that networks used to benchmark the algorithm are very small compared to even small scale state-of-the-art. Recently derived scaling laws (e.g Scaling Laws for Neural Language Models, https://arxiv.org/abs/2001.08361) were pointing to the fact that increasing model and data scale during training improves generalization and transfer - I therefore wonder, whether testing algorithms that aim on strong generalization across tasks with very small scale networks and data may lead to distorted picture, where a more sophisticated algorithm may be superior on small scale, but substantially underperform against simpler algorithms on larger scale.

Finally, one major unclear point to me is what authors attempt to elaborate on in discussion: how to organize and coordinate the contrastive learning which requires creation of the augmented version of the original task and careful execution of both original and augmented task learning in proximity to each other. I think authors attempt to bring in hippocampus as responsible structure falls short of explanation how it can manage to do those tasks - it feels to me like postulating a homunculus in form of hippocampus that has to figure out how to drive contrastive learning for all other brain networks involved. The authors should present at least some coarse explanation why they think that specifically hippocampus (and not, say, basal ganglia, prefrontal cortex, etc) is the candidate structure and how it may fulfull the required functions of creating augmented task and executing learning of both in coordinated way. Referring to "transfer" of "additional data" from hippocampus to cortex during sleep or resting state only adds to confusion here, in my opinion, as it introduces a very long time lag between original and augmented tasks and additional requirenment of specificity, where hippocampus should exactly to transfer data from a certain task to, and whether the target that should receive this data was not already exposed to data from various other tasks, which is not clear to me how to resolve.

---

> ### Author Response · Authors · 2022-08-02
> **Reply to reviewer YWLg part I**
>
> Thank you for your comprehensive remarks and questions. We address the points you raised one-by-one below.
>
> > 1. It seems that the rule relies on knoweldge of the task identity . It would be good to discuss how learning algorithm may obtain this knoweldge in biological plausible way and still remain local.
>
> The typical meta-learning setup we consider in this paper indeed assumes that the task identity is provided to the system. However, we would like to point out that prior work has developed approaches that can remedy this assumption. For example, the task identity can be treated as an unobserved random variable which is then inferred, as in the forget-me-not process [1] used in the elastic weight consolidation continual learning paper [2]. Such a complementary system in charge of determining task identity from the data can in principle be combined with meta-learning algorithms such as ours; we leave the investigation of biologically-plausible versions of task-identification systems to future work.
>
> Alternatively, the meta-objective can be redefined, removing the notion of tasks altogether, as proposed in recent investigations of task-agnostic meta-learning (e.g. [3]); our algorithm can be readily applied to this setting. Moreover, in this case, the meta-objective contains ‘replayed’ data from the past, which fits well with the view put forward in our discussion, where a complementary learning system which stores past experiences provides the data needed to evaluate the meta-objective. We will include a brief discussion of these points in the next version of our paper.
>
> [1] K. Milan, K. et al. (2016). The forget-me-not process. Advances in Neural Information Processing Systems
>
> [2] J. Kirkpatrick (2017). Overcoming catastrophic forgetting in neural networks. Proceedings of the national academy of sciences, 114(13), 3521-3526.
>
> [3] J. Rajasegaran et al. (2022). Fully online meta-learning without task boundaries. arXiv preprint arXiv:2202.00263.
>
> > 2. Does the training assume i.i.d shuffled data? If yes, how is that executed in biological plausible way? If no, what makes the algorithm perform continual learning?
>
> While the benchmarks we consider here perform gradient-based optimization on i.i.d. data, our meta-learning algorithm is agnostic to the concrete learning algorithm as long as it minimizes the loss. Learning from non-i.i.d. data is an exciting avenue to explore and in fact the synaptic and modulatory model we consider here have previously been suggested in meta-learning-based approaches for continual learning [4,5,6]. Our algorithm can be applied without modification to this setting (as a plug-in replacement for backpropagation through learning). These are interesting experiments to investigate in future work and we will discuss this point in the next version of our paper.
>
> [4] G. Gupta et al. (2020). Look-ahead meta learning for continual learning. Advances in Neural Information Processing Systems
>
> [5] M. Caccia et al. (2020). Online fast adaptation and knowledge accumulation (OSAKA): a new approach to continual learning. Advances in Neural Information Processing Systems
>
> [6] X. He et a. (2019). Task agnostic continual learning via meta learning. arXiv preprint arXiv:1906.05201.
>
> > 3. Why not combining both complex synapse and top-down modulation implementation in one rule? It seems both mechanisms are co-existing in the brain.
>
> This is an interesting idea. Following your suggestion, we have run an additional experiment on the wheel bandit task where we use the synaptic model to learn a consolidated state for the gain and adaptive threshold. Despite promising performance when validating the new model on the non-spiking sine wave regression task, we found that this model performed poorly on the wheel bandit task:
>
> | $\delta$                    | 0.5               | 0.9              | 0.99               |
> |-----------------------------|-------------------|------------------|--------------------|
> | CML (synaptic + modulatory) | 1.88$^{\pm 0.87}$ | 9.14$^{\pm 2.3}$ | 34.00$^{\pm 2.77}$ |
>
> To obtain this result we performed a comparable hyperparameter scan as for the other methods, sampling 1000 random configurations from the hyperparameter search space.

---

> > ### Author Response · Authors · 2022-08-02
> > **Reply to reviewer YWLg part II**
> >
> > > 4. What about benchmarking the algorithm on more realistic multi-task datasets? For instance, the standard ATARI game suit or MetaDataset (https://github.com/google-research/meta-dataset) would give a proper multi-task setting and allow for more generalizable comparision to other approaches.
> >
> > We agree that the choice of benchmark is important in the context of meta-learning. For this first presentation of our algorithm we have focused on a variety of established, yet challenging benchmarks that have been extensively studied by prior meta-learning algorithms to allow for a comprehensive comparison, in particular to previous gradient-based methods that are most relevant to our work. Nevertheless, scaling our method to more complex settings is an important goal for future work.
> >
> > > 5. In the discussion, you point to hippocampus as a "control" structure that organizes contrastive learning - creating augmented task version and coordinating original and augmented task learning. It would be good to give a more grounded explanation - why necessary hippocampus (and not another structure, say basal ganglia or pre-frontal cortex, or insula, that also have access to most of cortical networks) and how it may subserve these functions? How sleep related activity may be relevant here, given that there will be a large time lag between learning episodes and interference with all possible tasks that happened during this period of time?
> >
> > Thank you for raising an important open question that our work shares with complementary learning systems theory and other theories of contrastive learning. More work is indeed needed to establish if the hippocampus can play the role we suggest here. Despite the lack of conclusive evidence, we believe that the hippocampus is well positioned to organize contrastive learning as recent studies point to its role in replaying experience not only during sleep but also during awake states  [7, 8, 9]. We will discuss these points and stress the speculative nature of our proposal in the next version of our paper.
> >
> > [7] M. P. Karlsson & L. M. Frank (2009). Awake replay of remote experiences in the hippocampus. Nature Neuroscience, 12(7)
> >
> > [8] M. F. Carr et al. (2011). Hippocampal replay in the awake state: a potential substrate for memory consolidation and retrieval. Nature Neuroscience, 14(2),
> >
> > [9] D. J. Foster (2017). Replay comes of age. Annu. Rev. Neurosci, 40
> >
> > > 6. Paper mentions one intermediate state that has to be temporary buffered after original task learning was executed, until augmented task learning has been performed. Is this state local to the synapses? Where is it in the update equations (is it $\phi_{\theta,0, \tau}$)? It would be good to make this explicit for the reader. What is the biological plausible way to perform this short-term storage?
> >
> > Thank you for pointing out the missing reference to the buffered state. We are referring to $\hat{\phi}_{\theta,0, \tau}$ and will clarify this in the text. Indeed this state is local to the synapse for the synaptic model and could potentially be supported by short-term synaptic processes. Finding evidence for precise mechanisms supporting such storage is an important open question our method shares with other contrastive learning theories, e.g. the short-term storage of synapse-specific products of pre- and post-synaptic activity over phases required by classical contrastive Hebbian learning.
> >
> > This problem has for example been studied in the context of learning neural networks with equilibrium propagation. Ernoult et al. [7] have shown that learning can be made local in time by continuously updating the synaptic weights during the nudged phase ($\beta > 0$). The very same algorithm can be applied at the meta-level and would remove the need to store $\hat{\phi}_{\theta, 0, \tau}$.
> >
> > [7] Ernoult, M., Grollier, J., Querlioz, D., Bengio, Y., & Scellier, B. (2020). Equilibrium propagation with continual weight updates. arXiv preprint arXiv:2005.04168.
> >
> > > 7. It is stated that meta-gradient approximation error can be made arbitrary small by either running further learning or decreasing nudging factor .  What may be a biologically plausible way to tune the nudging factor?
> >
> >
> > The nudging factor $\beta$ is a global, scalar factor that could potentially be controlled through neuromodulation. As to how its value may be found, since beta is a scalar it can be treated like other hyperparameters (such as a learning rate) and be tuned by simple local search mechanisms (stochastic trial and error).

---

> > > ### Author Response · Authors · 2022-08-02
> > > **Reply to reviewer YWLg part III**
> > >
> > > > 8. It is mentioned that in case of implementing the rule by top-down modulation, the partial derivatives of the augmented loss function will be computed by automatic differentiation using standard backprop. To rescue biological plausibility, further mechanisms are invoked to replace backprop : prediction error neural subpopulations, dendritic error representations, or usage of equilibrium propagation. Would the same mechanisms then act for replacing backpropagation in general also for the original task learning, or is the idea here that those will be acting only for learning on augmented task?
> > >
> > > These methods provide a mechanism to solve a learning problem and would in the case of our algorithm be invoked twice, once for learning the task and once again for solving the augmented learning problem.
> > >
> > > > 9. In general, how would usual approaches to replace backpropagation with local rules (again, for instance equilibrium propagation) relate to the proposed contrastive rule - are they complementary, or belong to the same class of methods, only applied for meta-learning in this case?
> > >
> > > Usual approaches to replace backpropagation tackle the problem of solving a conventional (single-level) optimization problem. Here we consider the problem of meta-learning formalized as a bilevel optimization problem. Generally speaking, typical approaches to replace backpropagation with local rules are not trivially applicable to the bilevel optimization problem. An exception to this however is equilibrium propagation that has been stated in such generality which allowed us to develop the meta-learning algorithm presented in this work.

---

> > > > ### Comment · Reviewer_YWLg · 2022-08-09
> > > > **Response to rebuttal**
> > > >
> > > > I would like to thank the authors for the thorough and elaborate response to the reviews, I appreciate the effort authors took to address various concerns and to perform further experiments.
> > > >
> > > > After going through rebuttal, my impression is still that there is a lot of uncertainty to application of the proposed algorithm in multi-task setting at realistic scale.
> > > >
> > > > When arguing for possible mechanisms to treat multiple tasks while avoiding interference or catastrophic forgetting, authors refer to other works on continual on-line learning or task-agnostic meta-learning. Those are important and useful references, however they do not contribute to understanding how the proposed algorithm of contrastive meta-learning may go along with those various proposals, as to me the fusion seems to be rather non-trivial (although authors claim that one can just plug in contrastive meta-learning as objective and use it in the referred works, I do not see why this should work straight forward). Here, the authors should be more specific, choose one particular idea of task-agnostic continual learning and describe in detail convincingly how full algorithm might work and what parts of it would make sure that despite multiple tasks and no boundaries, algorithm would learn to perform both task inference and task solving without suffering classical problems (for instance, how would algorithm perform in face of dominance of one task type and only rare presentation of another? Think of playing pacman on ATARI most of the time, and showing pong and breakout only rarely - how the contrastive meta-learning would perform in such a scenario?)
> > > >
> > > > In my impression, authors did not really manage to address the issue with the demonstrations of the functionality of the proposed algorithm being small scale. It is often the case that increasing the scale completely changes the picture of comparing performance of different learning algorithms. Having a proper challenging multi-task benchmark setting (eg ATARI) would also help here a lot to clarify how well algorithm copes with various changing tasks. It is still to me unclear without such demostration how valuable the approach will be in the realm of more complex tasks than benchmarked in the current work.
> > > >
> > > > With regard to plugging in arguments of biological plausibility, I think authors still did not manage to go beyound vague hints on hippocampal replay, sleep and wake states, in arguing how contrastive meta-learning may have a neural implementation. Replay exists in different scenarios and across different structures in the brain (see for instance [1, 2]), so when arguing for specific implementation, one should be able at least to pinpoint what features of contrastive meta-learning are supported by the certain postulated neural mechanism and why other similar mechanisms are not suitable and can be ruled out. Without such detailed elaboration, biological plausibility is just a mere speculation that can be arbitrary wrong. While it is in my opinion not wrong to speculate about possible roles of reactivation in the cortical and subcortical structures as candidate mechanism for the proposed algorithm, it should be made clear that there is no specific candidate yet available if the authors cannot describe in detail how they imagine neural implementation would support the function of the algorithm.
> > > >
> > > > While I am convinced that paper is a relevant and interesting read for the community and therefore suggest acceptance, I will maintain my score.
> > > >
> > > >
> > > > [1] Rusu, S.I. and Pennartz, C.M., 2020. Learning, memory and consolidation mechanisms for behavioral control in hierarchically organized cortico‐basal ganglia systems. Hippocampus, 30(1), pp.73-98.
> > > >
> > > > [2] Oberto, V.J., Boucly, C.J., Gao, H., Todorova, R., Zugaro, M.B. and Wiener, S.I., 2022. Distributed cell assemblies spanning prefrontal cortex and striatum. Current Biology, 32(1), pp.1-13.

---

> > > > > ### Author Response · Authors · 2022-08-09
> > > > > **Reply to reviewer YWLg**
> > > > >
> > > > > We thank the reviewer for their response and regret that our reply has not convincingly addressed all of their concerns. We believe that the remaining concerns are exciting future research directions, but that they go beyond the scope of the current paper.
> > > > >
> > > > > In its current scope, our paper joins an active line of research that aims to understand the mechanisms which could support gradient computation in the brain [1]. This line of research has so far focused on learning. To the best of our knowledge, we are the first to study how gradient computation could be supported by the brain in the challenging meta-learning setting. Existing algorithms in this setting require going backwards in time and computing second-order derivatives, two considerable obstacles to envision meta-learning in the brain. We introduce a meta-learning rule that removes both obstacles and still performs competitively to prior algorithms on established meta-learning benchmarks opening the possibility to study our rule in large-scale problems.
> > > > >
> > > > > [1] Lillicrap, Timothy P., et al. "Backpropagation and the brain." Nature Reviews Neuroscience 21.6 (2020): 335-346.

---

### Official Review · Reviewer_rbQw · 2022-07-10

**Rating:** 6
**Confidence:** 3
**Soundness:** 3 good
**Presentation:** 3 good
**Contribution:** 3 good

**Summary:**

The authors propose a contrastive meta-learning rule having several appealing properties: (1) biologically-plausibility, (2) locality, (3) causality, and (4) learning process agnostic. Specifically, the rule contrasts the outcome of two learning episodes with/without meta-objective to update meta-parameters. The proposed meta-learning rule estimates meta-gradient by simply running the learning algorithm twice compared with the existing meta-learning methods which require either second-order terms or inverse learning loss Hessian. The authors theoretically analyze the meta-gradient estimation error and confirm this through the experiment results. Also, the proposed meta-learning rule shows similar performance to the existing meta-learning methods on various tasks and architectures.

**Questions:**

- One of the main contributions of this paper is its simplicity. However, it is not clearly written whether such simplicity leads to better computational and memory efficiencies than the existing meta-learning rules. Therefore, it would be great to add a comparison table for computational and memory complexity.
- In Fig. 2A, it looks like input and output are Poisson spike trains. However, since the input and output are continuous values, please redraw Fig. 2A.
- In Section 5.5, the modulatory model performs much worse than the synaptic model and MAML as increases a task-specific radius while it outperforms the synaptic model on visual few-shot learning. However, CAVIA [1], which uses a modulatory approach as cited in the paper, outperforms MAML in the RL Cheetah experiments. So, it would be great to see the performance of CAVIA on the wheel bandit problem.

References\
[1] Luisa Zintgraf et al. Fast context adaptation via meta-learning. In International Conference on Machine Learning, 2019.

**Limitations:**

Yes.

**Strengths And Weaknesses:**

Originality: The authors propose a novel method for meta-learning and applying this rule to the synaptic and modulatory models. The proposed method is simpler than the existing meta-learning methods including MAML and iMAML because it estimates the meta-gradients using only partial derivatives rather than second-order derivatives or inverse Hessian.

Quality: The authors' claims are well supported in the paper. The authors theoretically prove the meta-gradient estimation error of the proposed meta-learning rule in terms of the nudging strength and demonstrate it via the experimental results. Also, the proposed meta-learning rule matches or outperforms the reference methods on various benchmarks even its simplicity. However, it is not clearly written whether the simplicity of the proposed rule leads to better computational and memory efficiencies.

Clarity: The paper is generally well-written and easy to follow.

Significance: The results in the paper are important because the proposed meta-learning rule is simple but its estimation error is theoretically proven and the performance matches the reference algorithms on various tasks. Therefore, I expect this paper is likely to be used in meta-learning literature.

---

> ### Author Response · Authors · 2022-08-02
> **Reply to reviewer rbQw part I**
>
> Thank you for your positive review and instructive suggestions that have helped to improve the paper.
>
> > One of the main contributions of this paper is its simplicity. However, it is not clearly written whether such simplicity leads to better computational and memory efficiencies than the existing meta-learning rules. Therefore, it would be great to add a comparison table for computational and memory complexity.
>
> We agree that such a comparison is a valuable addition to the paper and we will add a respective table. The table highlights that CML is the only method that can approximate the meta-gradient as accurately as needed while only requiring first-order derivatives. As is also the case for implicit methods, its memory requirements are independent of the number of steps taken during the learning process.
>
> | Method   | # gradients w.r.t. |          | # 2nd-order terms |               | Memory                       | Exact in the limit |
> |----------|--------------------|----------|-------------------|---------------|------------------------------|-------|
> |          | $\phi$             | $\theta$ | HVP               | cross der. VP |                              |       |
> | CML      | T + K              | 2        | 0                 | 0             | $O(\|\phi\| + \|\theta\|)$   | ✓     |
> | CG       | T + 1              | 1        | K                 | 1             | $O(\|\phi\| + \|\theta\|)$   | ✓     |
> | RBP      | T + 1              | 1        | K                 | 1             | $O(\|\phi\| + \|\theta\|)$   | ✓     |
> | T1-T2    | T + 1              | 1        | 0                 | 1             | $O(\|\phi\| + \|\theta\|)$   | ✗     |
> | BPTL     | T + 1              | T + 1    | T                 | 0             | $O(T \|\phi\| + \|\theta\|)$ | ✓     |
> | TBPTL    | T + 1              | K + 1    | K                 | 0             | $O(K \|\phi\| + \|\theta\|)$ | ✗     |
> | FOBPTL   | T + 1              | T + 1    | 0                 | 0             | $O(T \|\phi\| + \|\theta\|)$ | ✗     |
> | Reptile* | T                  | 0        | 0                 | 0             | $O(\|\phi\|)$                | ✗     |
>
> _Caption_: Comparison of computational and memory complexity of different meta-learning methods. T denotes the number of steps in the base learning process and K refers to steps taken in an algorithm-specific second phase. “HVP” abbreviates “Hessian-vector product” and “cross der. VP” denotes “cross derivative vector product”. The algorithms compared in this table are contrastive meta-learning (CML), conjugate gradients (CG; used in iMAML), recurrent backpropagation (RBP), T1-T2, backpropagation through learning (BPTL; used in MAML), its truncated version (TBPTL) and its first-order version (FOBPTL; all Hessians are replaced by the identity, also known as FOMAML) and Reptile. * Reptile is not a general-purpose meta-learning method as it is restricted to meta-learn the initialization of the learning process.
>
> > In Fig. 2A, it looks like input and output are Poisson spike trains. However, since the input and output are continuous values, please redraw Fig. 2A.
>
> Thank you for pointing out this typo. While the inputs are indeed encoded as spikes before being input to the RSNN, this is not the case for the output unit, which is as you point out continuous rather than spiking. We will correct Fig. 2A accordingly and adapt the text to describe the setup more carefully.

---

> > ### Author Response · Authors · 2022-08-02
> > **Reply to reviewer rbQw part II**
> >
> > > In Section 5.5, the modulatory model performs much worse than the synaptic model and MAML as increases a task-specific radius while it outperforms the synaptic model on visual few-shot learning. However, CAVIA [1], which uses a modulatory approach as cited in the paper, outperforms MAML in the RL Cheetah experiments. So, it would be great to see the performance of CAVIA on the wheel bandit problem.
> >
> > This is indeed a very relevant experiment that we have now conducted following your suggestion. We adopted the exact setup used by CAVIA for the RL Cheetah experiments which uses a multilayer perceptron of the same size as we are using for the other methods (2 hidden layers with 100 hidden units each) and a context embedding of size 50 that is concatenated to the inputs and adapted during the learning process. Despite an extensive hyperparameter scan, we found that CAVIA performs worse than the other meta-learning methods we tested on the wheel bandit task:
> >
> > | $\delta$ | 0.5               | 0.9               | 0.99               |
> > |----------|-------------------|-------------------|--------------------|
> > | CAVIA    | 1.21$^{\pm 0.04}$ | 4.29$^{\pm 0.25}$ | 23.99$^{\pm 1.68}$ |
> >
> > In order to validate our implementation of CAVIA, we ran the sine wave regression task in section 5.1.1. of [1] where we found comparable results.
> > The random hyperparameter search we ran evaluated 1000 random samples from the hyperparameter search space as for the other methods.
> >
> > [1] Luisa Zintgraf et al. Fast context adaptation via meta-learning. In International Conference on Machine Learning, 2019.

---

### Official Review · Reviewer_6moM · 2022-07-11

**Rating:** 7
**Confidence:** 3
**Soundness:** 3 good
**Presentation:** 4 excellent
**Contribution:** 3 good

**Summary:**

The authors proposed a biologically inspired approach to meta-learning that avoids backpropagation through time. They change the optimization process and reorganize the loss function such that it mixes problem-specific and meta-specific parameters.

**Questions:**

It would be helpful if the authors discuss how their approach would scale to more complex problems.

**Limitations:**

I believe a discussion about the limitations of the experimental results would help to better understand the strengths and weaknesses of the proposed approach. For example, the authors could discuss the possibility of using this method on more challenging meta-learning RL tasks.

**Strengths And Weaknesses:**

The paper provides a novel approach to meta-learning. The new approach is more biologically plausible than existing approaches: it is time local and does not require backpropagation through time. The paper is well written and the authors conducted an informative theoretical analysis of their approach.

The authors conducted experiments on several different types of datasets. They obtained better results than SOTA on meta-learning CIFAR-10, but comparable results on other datasets. While I think that the paper would be stronger with more experiments on more challenging datasets, I believe it is still a solid contribution.

---

> ### Author Response · Authors · 2022-08-02
> **Reply to reviewer 6moM**
>
> Thank you for your comments and for taking the time to review our work. Below we reply to your question on how our approach scales to more complex problems and discuss in more detail the limitations of the experimental results.
>
> > It would be helpful if the authors discuss how their approach would scale to more complex problems.
>
> Thank you for the suggestion, scaling our approach to more complex problems is indeed an exciting next step and calls for further discussion. Compared to other meta-learning methods, CML is simple to implement and has a small additional memory footprint. This potentially allows meta-learning bigger models than competing algorithms which are often limited by memory constraints. We will add a respective comparison of the computational and memory complexity of different meta-learning methods to the paper to highlight this point, [please see the response to reviewer rbQw here](https://openreview.net/forum?id=NIJFp_n4MXt&noteId=J9rxKfdwRZN).
>
> When moving to larger model architectures and more difficult tasks, we believe that one of the main challenges will be to ensure that the learning process comes close to a solution of the learning problem, as required both by CML and implicit methods. A potential remedy is to smartly pick architectures where only few parameters are adapted in the learning process (such as exemplified by our modulatory model) or to design an architecture that affords an analytical solution to the learning process (such as a linear readout, e.g. [1], [2]) and treat the majority of parameters as meta-parameters.
>
> [1] Lee, et al. "Meta-learning with differentiable convex optimization." Proceedings of the IEEE/CVF conference on computer vision and pattern recognition. 2019.
>
> [2] Bertinetto, et al. "Meta-learning with differentiable closed-form solvers." International Conference on Learning Representations. 2019.
>
> > I believe a discussion about the limitations of the experimental results would help to better understand the strengths and weaknesses of the proposed approach.
>
> We agree that the limitations of the experimental results should be more carefully discussed and we will adapt the discussion accordingly. Specifically, both implicit methods and CML assume that the learning problem is solved to completion. This requirement is both a strength of the method as we can go beyond the few update steps regime of MAML-like methods but it also implies that we cannot directly meta-learn meta-parameters that only impact the learning process but not the final solution, such as learning rates. This limitation holds for any other implicit method. In practice, we can potentially augment the loss or architecture to achieve similar effects, for example, the attraction strength $\lambda$ of our synaptic model can be thought of as a way to modulate the learning rate.

---

### Official Review · Reviewer_1nAt · 2022-07-11

**Rating:** 8
**Confidence:** 3
**Soundness:** 4 excellent
**Presentation:** 4 excellent
**Contribution:** 4 excellent

**Summary:**

The authors propose a novel meta-learning algorithm, effectively applying Equilibrium Propagation (EP, Scellier and Bengio, 2017) to the dynamical system defined by a learning algorithm minimizing a loss function (rather than the dynamical system defined by a model minimizing an energy function as in the original EP work). The authors define an augmented loss function (analogous to the "total energy function" of EP) consisting of a learning loss and an evaluation loss, weighted by a nudging strength. To estimate the total derivative with respect to the meta-parameters, they take the difference of the partial derivatives of the nudged and non-nudged loss, thus making a finite-difference approximation of the total derivative. Finally, the authors validate the algorithm with two specific meta-learning models (synaptic and neuronal) applied to four tasks (hyperparameter tuning, few-shot learning, regression, and reinforcement learning), demonstrating competitive performance, and provide a biological interpretation of their models.

**Questions:**

- Are there cases where the meta-learning fails (e.g. does not converge to a viable solution for some/all of the learning tasks) due to the approximation? It would be instructive to include an example where CML entirely fails where another algorithm succeeds and vice versa (if such examples exist).
- In section 5.2, the authors compare two methods (RBP, CG) which incur numerical errors in calculating the meta-gradient with CML's approximation. Is it possible to include an "upper-bound" result using the exact meta-gradient (computed with BPTL, or analytically as in section S3.6)?
- In section 5.3, MAML perhaps plays the role of BPTL as an "upper bound" for the performance. However, it is difficult to compare these side-by-side as it is unclear whether the performance differences result from a superior meta-learning algorithm, or a superior learning algorithm, or differences in architectures (number of tuneable parameters, number of tuneable meta-parameters).
- In section 5.4, can you clarify how the inputs are presented to the network, and how the outputs are read out? (e.g. for a given sinusoid amplitude/phase, is x_i a spike with probability p_i? Is y_k averaged across all 20 time steps? Is there only one y_k?)
- In section 5.4, it is surprising that the BPTT+eprop performance is much worse than CML+eprop. If CML is an approximation of the exact meta-gradient from BPTT, why isn't BPTT strictly better?
- Potential reference to include for section 4.2 or discussion: Titley et al., Toward a neurocentric view of learning, Neuron 2017


**Limitations:**

Technical limitations partially addressed through analysis of estimation error (see also Questions). Societal impact not discussed (but does not seem necessary for this work).

**Strengths And Weaknesses:**

Originality. The paper is an interesting application of Equilibrium Propagation to meta-learning, suggesting a biological implementation/approximation of backpropagation-through-learning (BPTL), analogous to how EP proposes a biological implementation of backpropagation-through-time. Although the contrastive rule itself is not novel, it is used in a different setting and its effectiveness there is thoroughly demonstrated. Furthermore, the authors extend the original theoretical analysis to include the approximation error incurred from the finite-difference step (i.e. \beta) as well as from non-convergence to the fixed point.

Quality. Overall, the paper is very thorough and rigorous, both theoretically with numerical validation of the theoretical results, and empirically in comparison to previous meta-learning benchmarks. A potentiall weakness is that for some of the empirical results on the various meta-learning tasks, it is unclear how to make a fair comparison to other methods (see Questions), equalizing architecture, computational resources, etc.

Clarity. Paper is very clearly written and well-organized.

Significance. From a machine learning perspective, this work provides a useful meta-learning approach which avoids storing the computational graph of the learning algorithm, thus using additional memory proportional to the number of meta-parameters (rather than the number of learning steps). The time complexity remains the same as BPTL. From a neurobiological perspective, it is an important step towards a biologically plausible meta-learning algorithm. Furthermore, the success of the modulatory CML algorithm is particularly interesting as it suggests a much more important role for learning via neuronal (rather than synaptic) changes than is typically considered.

---

> ### Author Response · Authors · 2022-08-02
> **Reply to reviewer 1nAt part I**
>
> Thank you for the encouraging review and for the useful feedback that we incorporated in a revised version of our paper. We reply to your questions point-by-point below.
>
> > Are there cases where the meta-learning fails (e.g. does not converge to a viable solution for some/all of the learning tasks) due to the approximation? It would be instructive to include an example where CML entirely fails where another algorithm succeeds and vice versa (if such examples exist).
>
> We have not encountered problems that can be formulated as Eq. 1 but not solved by CML given sufficient compute time and an adequate choice of hyperparameters. However, if the former is not the case and the solution to the two learning problems is not close enough to a (local) minimum, CML can fail to extract a sufficiently good meta-gradient signal.
> Such a case can be observed in Figure 1B when the number of first phase steps is small. The validation loss after meta-learning deteriorates even for an appropriately chosen value for $\beta$ the fewer number of first phase steps are used. CML shares this failure mode with other implicit gradient-based meta-learning methods and we have not found examples in which CML fails but not other implicit methods.
>
> Additionally, not all meta-learning problems can be formulated with Eq. 1 because of the assumption that the learning loss is minimized. Algorithms based on backpropagation through learning do not require this and allow meta-learning of systems that take only a few learning steps or do not minimize any loss. They typically work well in the opposite regime of few learning steps but potentially suffer for longer learning trajectories due to vanishing/exploding gradients and large memory requirements.
>
> > In section 5.2, the authors compare two methods (RBP, CG) which incur numerical errors in calculating the meta-gradient with CML's approximation. Is it possible to include an "upper-bound" result using the exact meta-gradient (computed with BPTL, or analytically as in section S3.6)?
>
> We agree that an "upper-bound" comparison would be useful to better contextualize the results in section 5.2. Unfortunately, the analytical solution to the meta-gradient is intractable in this case because of the inverse Hessian term (cf. Eq. 25) and running full BPTL requires too much memory. We can however run truncated BPTL, choosing the truncation length such that we fully use the memory available to us. We managed to backpropagate through the last 200 steps of the 5000 step long learning process, resulting in an evaluation accuracy of 64.99$^{\pm 0.39}$ and a test accuracy of 73.93$^{\pm 0.63}. This result helps to reinforce that CML is a strong meta-optimizer, even outperforming TBPTL in its ability to fit the meta-parameters to the validation set, as it reaches greater accuracy on the evaluation data. Interestingly, CML’s strong meta-optimization seems to enter a regime where “meta-overfitting” occurs as the solution found by TBPTL generalizes better to the test set. We specifically did not put in place any measures against “meta-overfitting” in this experiment (such as for example “meta-regularization”) as we are interested in assessing the meta-optimization capabilities of the various methods. We will include these results in the final version of the paper.
>
> | Method | Evaluation acc. (%)  | Test acc. (%)      |
> |--------|----------------------|--------------------|
> | TBPTL  | 64.99$^{\pm 0.39}$   | 73.93$^{\pm 0.63}$  |
> | T1-T2  | 64.77$^{\pm 0.40}$   | 62.57$^{\pm 0.31}$ |
> | CG     | 57.65$^{\pm 1.51}$   | 57.51$^{\pm 0.98}$ |
> | RBP    | 64.92$^{\pm 1.32}$   | 62.14$^{\pm 0.97}$ |
> | CML    | 74.43$^{\pm 0.53}$   | 66.94$^{\pm 0.25}$ |
>
> > In section 5.3, MAML perhaps plays the role of BPTL as an "upper bound" for the performance. However, it is difficult to compare these side-by-side as it is unclear whether the performance differences result from a superior meta-learning algorithm, or a superior learning algorithm, or differences in architectures (number of tuneable parameters, number of tuneable meta-parameters).
>
> The comparison in this experiment is indeed rendered difficult given that the number of learning steps and learning algorithm depends on the particular meta-learning algorithm. To make the comparison as fair as possible, all methods share the same base network. Furthermore, MAML, FOMAML, Reptile and iMAML have the same number of trainable meta-parameters as does CML with the synaptic model which only meta-learns the consolidated state $\omega$ in this case. The most immediate comparison is possible between iMAML and CML with the synaptic model as both methods run for many learning steps and meta-learn the same number of meta-parameters. We will clarify this point in the main text.

---

> > ### Author Response · Authors · 2022-08-02
> > **Reply to reviewer 1nAt part II**
> >
> > > In section 5.4, can you clarify how the inputs are presented to the network, and how the outputs are read out? (e.g. for a given sinusoid amplitude/phase, is x_i a spike with probability p_i? Is y_k averaged across all 20 time steps? Is there only one y_k?)
> >
> > Thank you for pointing out that our description here is incomplete, we will extend our description in the paper accordingly.
> > To clarify, we normalize the scalar input $x \in [-5,5]$ to the range $z \in [0,1]$ and encode it with a population of 100 neurons, each of which has a Gaussian response field with the mean values $\mu_i$ evenly distributed from 0 to 1 and a fixed variance $\sigma^2$. For a given input, each neuron thus has a firing probability of $p_i = \exp(\frac{- (\mu_i - z)^2}{ 2 \sigma^2})$ from which we sample a spike for each of the 20 time steps as inputs to the RSNN.
> > The output of the RSNN is non-spiking. It is the average potential of the single readout unit over all 20 time steps which we compare to a single scalar target value when computing the loss. As pointed out by reviewer rbQw, this is not reflected in Figure 2A and we will change it accordingly.
> >
> >  > In section 5.4, it is surprising that the BPTT+eprop performance is much worse than CML+eprop. If CML is an approximation of the exact meta-gradient from BPTT, why isn't BPTT strictly better?
> >
> > This is indeed a surprising observation. We carefully tuned the hyperparameters again but despite a slight improvement (validation MSE 0.52$^{\pm 0.05}$, test MSE 0.72$^{\pm 0.08}$) the qualitative finding remains. We hypothesize that the discrepancy is due to the limited number of gradient steps we can afford when backpropagating through the eprop learning process. Potentially, eprop disproportionately benefits from more learning steps as can be afforded by CML. In order to shed further light on this phenomenon we ran an additional experiment using truncated backpropagation through the eprop learning process where we run the learning process for the same number of steps as CML (500) but limit backpropagation through learning to the last 100 update steps such that it still fits in GPU memory in our setup. In line with the stated hypothesis, this experiment yields slightly improved numbers, i.e. validation MSE: 0.27 $^{\pm 0.07}$ and test MSE: 0.50 $^{\pm 0.11}$. We will add this comparison as well as the updated numbers to Tab. 4.
> >
> > > Potential reference to include for section 4.2 or discussion: Titley et al., Toward a neurocentric view of learning, Neuron 2017
> >
> > Thank you for suggesting this relevant reference. We were indeed not aware of it and will add it to the discussion of the modulatory model.

---

### Author Response · Authors · 2022-08-02
**Summary of replies to all reviewers**

We thank all reviewers for their constructive criticism and useful suggestions that have helped us improve our paper. We provide below a summary of the major changes made in response to the reviews:

-   We added the truncated backpropagation through learning (TBPTL) baseline for the CIFAR-10 hyperparameter optimization experiments of Section 5.2, as suggested by Reviewer 1nAt ([see here](https://openreview.net/forum?id=NIJFp_n4MXt&noteId=sLEjTCmtiO3)). The goal of this experiment is to better contextualize the existing results providing a comparison that matches full backpropagation through learning (BPTL) as closely as possible given memory constraints. It confirms that CML is a powerful meta-optimizer.

-   Reviewer 1nAt pointed out the surprising inefficiency of the BPTL-eprop baseline in the spiking sine wave regression experiments of Section 5.4. We hypothesized that eprop suffers more than backpropagation through time (BPTT) from the short learning process imposed by the memory constraints of BPTL. This hypothesis is supported by the following experiment: we let the eprop learning algorithm run until equilibrium but truncated the backward pass so that the computational graph fits in memory, and observed that the evaluation loss noticeably improves ([see here](https://openreview.net/forum?id=NIJFp_n4MXt&noteId=8y567eNKMF6)).

-   As suggested by Reviewer rbQw, we summarized the computational and memory complexity of CML and compared it to other meta-learning methods ([see here](https://openreview.net/forum?id=NIJFp_n4MXt&noteId=J9rxKfdwRZN)). Notably, the memory footprint of CML (and of other implicit methods) is much smaller than methods based on BPTL, and CML is the only algorithm that allows approximating the meta-gradient as accurately as needed while only using first-order derivatives. These properties make CML attractive for scaling up to more complex datasets.

-   We will fix a typo in Figure 2A noticed by reviewer rbQw and illustrate the output unit with a continuous-valued output in line with our experimental setup of section 5.4.

-   As suggested by reviewer rbQW, we evaluated CAVIA [1] on the reward-based learning experiment of section 5.5. Despite promising performance on a debugging task and a thorough hyperparameter scan, we found CAVIA to perform worse than the other meta-learning methods ([see here](https://openreview.net/forum?id=NIJFp_n4MXt&noteId=Dhn3YdEndaf)).


[1] Luisa Zintgraf et al. Fast context adaptation via meta-learning. In International Conference on Machine Learning, 2019.

---

### Author Response · Authors · 2022-08-09
**Still available for discussion**

Dear reviewers,

thank you once again for your useful comments. The end of the author-reviewer discussion period is coming close, and we see that most of you have not yet updated your reviews in response to our new results and proposed changes addressing your concerns. In case something is unclear, please do not hesitate in contacting us before the interactive discussion period ends.

---

### Meta-Review · Area_Chair_YXxv · 2022-08-27

**Recommendation:** Accept
**Confidence:** Certain

**Metareview:**

The Reviewers appreciated the novelty factor of the contrastive meta-learning algorithm proposed in the paper, the theoretical analysis establishing a formal connection with equilibrium propagation, and the appealing features of the resulting meta-learning procedure, which include memory and computation efficiency, as well as the fact that the algorithm affords a biologically-plausible implementation that only requires locally available information for parameter updates.
To concretely showcase these properties the paper demonstrates two instantiations of the proposed algorithm that are mechanistically realized through synaptic consolidation and top-down neuronal modulation, respectively.
Finally, the paper validates the algorithm on standard few-shot learning benchmarks.
The main weaknesses of the paper identified by the Reviewers are the empirical evaluation, which would benefit from a more extensive comparison between methods and more experiments on more challenging meta-learning datasets, and the discussion on the candidate neurobiological substrate for a brain implementation of contrastive meta-learning, which would benefit from a more detailed and systematic description.
These limitations however do not substantially detract from the overall quality, relevance and interest of the paper, which Reviewers unanimously recommend for acceptance.

**Award:**

No

---

### Decision · Program_Chairs · 2022-09-14

Accept